# Nested Spatio-Temporal Time Series Forecasting

Yinghao Ai[*◇1 2 4]   Yukai Zhou[*◇2 3]   Ruoxi Jiang[†2 3]   Junyi An[†2]   Chao Qu[†2 3]   Zhijian Zhou[2 3]   Shiyu Wang[6]
Fenglei Cao[2]   Zenglin Xu[†2 3]   Furao Shen[†1 5]   Yuan Qi[†2 3]

## Abstract

Spatiotemporal forecasting is critical for real-world applications like traffic management, yet capturing reliable interactions remains challenging under noisy and non-stationary conditions. Existing methods primarily rely on historical spatial priors, often failing to account for evolving temporal correlations and suffering from systematic errors. In this work, we propose a nested forecasting framework that couples future macro-level regional trends with micro-level historical observations, enabling top-down guidance from abstract future representations for fine-grained forecasting. Specifically, we employ a spectral clustering-based approach to construct semantically coherent regions, providing both theoretical and empirical evidence that this representation effectively filters systematic noise while preserving essential trends. Building on this, we develop a progressive coarse-to-fine predictor to integrate these representative features into the inference process. This enables the model to leverage trend predictions to anticipate dynamic anomalies, such as periodic offsets, in advance. Furthermore, extensive experiments on multiple high-dimensional datasets demonstrate that our method consistently outperforms state-of-the-art baselines, validating the effectiveness of future macro-guided nested forecasting.

---

[*]Equal contribution   [◇] Work done during the internship at Shanghai Academy of AI for Science.   [†] Corresponding authors. [1]National Key Laboratory for Novel Software Technology, Nanjing University, China [2]Shanghai Academy of AI for Science, Shanghai [3]Fudan University, Shanghai [4]Department of Computer Science and Technology, Nanjing University [5]School of Artificial Intelligence, Nanjing University [6]ByteDance . Correspondence to: Ruoxi Jiang <roxie_jiang@fudan.edu.cn>, Furao Shen <fr-shen@nju.edu.cn>.

*Proceedings of the 43rd International Conference on Machine Learning*, Seoul, South Korea. PMLR 306, 2026. Copyright 2026 by the author(s).

## 1. Introduction

Spatio-temporal forecasting (STF) plays a pivotal role in modern intelligent systems, supporting diverse applications from urban traffic management to extreme weather prediction (Jin et al., 2024; Kumar et al., 2025; Wang et al., 2025a; Chen et al., 2025; Wang et al., 2025b). In these domains, accurate forecasting is indispensable for proactive decision-making, such as early congestion control (Hamed-moghadam et al., 2022) and emergency planning (Guo et al., 2023). Nevertheless, achieving robust predictions over extended horizons remains a significant challenge, primarily due to the complex spatial interactions and dynamic temporal patterns inherent in real-world systems (Lan et al., 2022; Chen et al., 2024; He et al., 2025; Gao et al., 2025).

As a specialized task within Multivariate Time Series (MTS) forecasting, STF focuses on improving prediction accuracy by effectively modeling spatial correlations (Shao et al., 2025). Early approaches (Li et al., 2018; Yu et al., 2018) typically integrated prior topological structures directly into graph-based learners; however, these priors often require expert knowledge, and might overlook the intricate patterns in the dynamic feature space. To better align with the inductive biases of dynamic data, subsequent methods (Wu et al., 2019; 2020b) introduced learnable adjacency matrices to adaptively capture latent edges with graph neural networks. Other research (Shao et al., 2022a; Jiang et al., 2023a; Diao et al., 2024; Ma et al., 2025a) has extended this to the temporal dimension, employing time-varying and multi-view graphs to model evolving interactions. Despite these architectural advancements, existing frameworks face a critical limitation: the fine-grained, full-graph modeling prevalent in current methods is highly susceptible to system noise, which becomes particularly acute with the growth of spatial scale. Within this expanded search space, models are prone to learn spurious correlations (Zhao et al., 2023), ultimately degrading the robustness of the learned representations.

To mitigate the impact of structural uncertainties and achieve robust forecasting, we investigate the utility of macro-level representations. A standard practice for constructing such representations involves the spatial aggregation or slicing; however, existing literature (Ma et al., 2023; Zhang et al., 2024; Fang et al., 2025) primarily treats these coarse-grained

signals as auxiliary inputs to provide robust historical statistics. In this work, we aim to explore a more promising paradigm: leveraging coarse-grained representations to characterize future states, thereby serving as a stable structural guide for the forecasting process. This approach introduces a significant challenge: how to extract coarse-grained signals with high representational fidelity while maintaining topological and semantic alignment with their fine-grained counterparts.

Motivated by these insights, we propose NEST(**Ne**sted **S**patio-**T**emporal forecasting), a spatio-temporal forecasting framework that moves beyond micro-level modeling through macro-guided design with future awareness. NEST realizes the cross-horizon modeling of historical fine-grained and future coarse-grained dynamics through two core stages. First, to extract representative macro-dynamics from local observations, we employ semantic spectral clustering (Ng et al., 2001), operating on an affinity matrix constructed directly from raw feature sequences, adapting to dynamic semantic correlations without relying on static physical priors while yielding a compact representation space. Second, to enable effective multi-scale interaction for temporal forecasting, we introduce a symmetric attention mechanism that facilitates bidirectional information flow across spatio-temporal scales. In particular, macro-level states are predicted multiple steps ahead, providing abstract future context that regularizes fine-grained forecasting. Crucially, the low-rank nature of the macro-state representation significantly reduces computational cost while preserving trend-level expressiveness. Our contributions are summarized as follows:

- We propose NEST, a nested spatio-temporal forecasting framework that introduces macro-guided cross-horizon modeling, using predicted region-level futures as explicit top-down guidance to regularize and enhance fine-grained forecasting.

- We design a computationally efficient multi-scale architecture that leverages semantic spectral clustering for capturing dynamic representative features, enabling robust alignment while better preserving systematic trends.

- We validate the effectiveness of NEST through extensive experiments on multiple large-scale datasets, achieving consistent improvements over state-of-the-art methods across diverse metrics.

## 2. Related Works

**Spatio-Temporal Forecasting**. The core objective of spatio-temporal forecasting is to predict future system states by capturing complex dependencies present in historical observations. Early works combined recurrent or temporal convolution modules with graph encoders that use fixed topologies to model spatial correlations (Li et al., 2018; Yu et al., 2018). To relax the reliance on predefined structures, later methods such as GWNET, MTGNN, and AGCRN introduced adaptive embeddings to infer latent spatial dependencies directly from observations (Wu et al., 2019; 2020b; Bai et al., 2020). Although these data-driven methods relax the reliance on predefined graphs by learning spatial relations from data, the inferred relational structures are typically static during inference. As a result, they still struggle to capture spatial dependencies that evolve over time, which are common in complex real-world systems (Han et al., 2021). To capture time-varying connectivity, a recent line of work leverages attention and dynamic message-passing mechanisms that adapt relations over time; representative examples include DSTAGNN, MEGACRN and STAEFORMER, among others (Lan et al., 2022; Jiang et al., 2023b; Liu et al., 2023a; Xie et al., 2023; Kong et al., 2024; Gong et al., 2024; Li et al., 2025). These approaches enable models to track changing dependencies and better handle nonstationary spatio-temporal dynamics (Lyu et al., 2025). Building on these dynamic capabilities, recent research further targets systemic challenges such as spatial heterogeneity (Ji et al., 2023; Dong et al., 2024) and the scalability of massive networks (Yuan et al., 2024; Fang et al., 2025). Despite these advances, most existing architectures remain focused on micro-scale modeling, remaining sensitive to noise and short-term irregularities.

**Hierarchical Spatio-Temporal Modeling.** Hierarchical structures help capture multi-scale spatio-temporal dynamics (Mao et al., 2025). Early works such as HGCN and HRNR grouped sensors into static regions to summarize macro-scale patterns (Guo et al., 2021; Wu et al., 2020a), and later extensions (e.g., HSDGNN) modeled more complex structural dependencies (Zhou et al., 2025). More recent methods introduce flexible cross-scale interactions: HISTGNN, HIEST and AIMST propose dynamic mechanisms to exchange information between sensor-level and region-level representations, while HSTAN and HSFE apply multi-level attention and feature fusion to integrate global and local correlations (Ma et al., 2022; 2023; Zhang et al., 2024; Marisca et al., 2024).

Beyond spatial hierarchies, several works explore temporal multi-scale modeling (Challu et al., 2023; Wang et al., 2023; Chen et al., 2023; Wang et al., 2024a). However, most prior methods treat hierarchy primarily as a mechanism for historical representations. They typically employ a single-stage projection that maps past observations directly, forcing the model to implicitly infer evolving trends from noisy data, making predictions highly susceptible to input perturbations. On the other hand, a recent work on neural operator (Jiang et al., 2025) demonstrates that incorporating

future information can improve long-term stability and physical consistency. However, its formulation assumes regular spatiotemporal grids and is less suitable for irregular graph-structured data with missing values and high noise. In this work, NEST establishes a hierarchical forecasting paradigm that explicitly predicts future macro-level states as structural guidance. By leveraging representative future dynamics to guide fine-grained generation, NEST stabilizes the forecasting process and alleviates the limitations of single-stage prediction.

## 3. Preliminary

We consider a spatiotemporal system consisting of $N$ correlated sensors, where observations at time $t$ are denoted by $\mathbf{X}_t \in \mathbb{R}^{N \times C}$. Our goal is to forecast a future sequence of length $H$ given a historical context of length $L$, denoted as $\mathbf{X}_{t-L+1:t} \in \mathbb{R}^{N \times L \times C}$.

To effectively model long-range dependencies while maintaining computational efficiency, we frame the problem as a patch-based autoregressive forecasting task. At each step, the model predicts a subsequent future patch of length $P$ from the preceding $L$ steps. During training, the model is optimized via single-step supervision on the predicted patch $\hat{\mathbf{X}}_{t+1:t+P}$. During inference, the full horizon $H$ is generated auto-regressively: the model consumes its own predicted patches as context for subsequent iterations until the entire sequence is realized.

## 4. Method

In this section, we present the NEST (Nested Spatio-Temporal) framework. NEST adopts a hierarchical coarse-to-fine paradigm, where fine-grained node-level predictions are guided by stable, macroscopic centroid dynamics to mitigate the impact of localized noise.

### 4.1. From raw data to centroid features

Direct node-level forecasting is challenging due to the high dimensionality of the output space, as well as the presence of local noise, missing values, and short-term irregular fluctuations. To address this, we leverage spectral clustering (Ng et al., 2001; Shi & Malik, 2000) to extract latent region-level representations, that serve as structural anchors for the system, enabling abstract guidance for fine-grained node-level forecasting.

**Constructing the Temporal Affinity Matrix.** An effective regionalization should reflect temporal coherence, meaning that nodes assigned to the same region exhibit consistent long-term co-movement patterns. Affinity matrices derived from physical proximity or predefined topology are often insufficient, as they fail to capture latent semantic

relationships that evolve with the system dynamics.

To address this limitation, we construct a feature-driven affinity matrix $\mathbf{A} \in \mathbb{R}^{N \times N}$ directly from raw temporal observations. Specifically, we partition the training sequence into $\tilde{T}$ non-overlapping temporal chunks, where $\tilde{T}$ is chosen to align with the intrinsic periodicity of the data (see Appendix A.7). For each node, we compute an averaged representation within each chunk and define pairwise affinities as

$$\mathbf{A}_{ij} = \exp\left( -\frac{1}{2\sigma^2 \tilde{T}} \sum_{k=1}^{\tilde{T}} \left\| \mathbf{X}_i^{(k)} - \mathbf{X}_j^{(k)} \right\|_2^2 \right), \quad (1)$$

where $\mathbf{X}_i^{(k)} \in \mathbb{R}^{T_k}$ denotes the temporal subsequence of node $i$ in chunk $k$, and $\sigma$ controls the kernel bandwidth. This construction emphasizes similarity in long-term temporal evolution rather than short-term fluctuations.

**Spectral representation.** Given the spatiotemporal affinity matrix $\mathbf{A}$, we compute the normalized graph Laplacian

$$\mathbf{L}_{\text{sym}} = \mathbf{I} - \mathbf{D}^{-\frac{1}{2}} \mathbf{A} \mathbf{D}^{-\frac{1}{2}}, \quad (2)$$

where $D$ is the degree matrix with $D_{ii} = \sum_j \mathbf{A}_{ij}$. The Laplacian characterizes how node features vary over the learned affinity graph: nodes with strong affinities are encouraged to have similar embeddings, while weakly connected nodes are allowed to diverge. In particular, the low-frequency eigenvectors of $\mathbf{L}_{\text{sym}}$ capture dominant and globally consistent correlation structures in the spatiotemporal system. These components provide a principled basis for identifying coherent latent regions aligned by long-range temporal behavior.

We then obtain $M$ regions with $M < N$ by applying K-means clustering to the spectral embeddings, resulting in an assignment matrix $S \in \{0, 1\}^{N \times M}$. This yields an assignment matrix $\mathbf{S} \in \{0, 1\}^{N \times M}$, where $S_{i,m} = 1$ indicates that node $i$ is assigned to region $m$. The region-level representation at time $t$ is computed via average pooling:

$$\mathbf{Z}_{t,m} = \frac{\sum_{i=1}^{N} S_{i,m} \mathbf{X}_{t,i}}{\sum_{i=1}^{N} S_{i,m}}, \quad m = 1, \ldots, M, \quad (3)$$

where $\mathbf{X}_{t,i} \in \mathbb{R}^C$ denotes the feature of node $i$ at time $t$. This operation compresses node-level observations into compact centroid features for hierarchical forecasting. Crucially, this aggregation acts as a natural low-pass filter that smooths out high-frequency local anomalies while preserving underlying regional trends. A visual demonstration of how these centroid features provide stable anchors compared to noisy raw sequences is provided in Appendix A.14.

To formally ground this intuition, assuming independent Gaussian noise $\epsilon_i$, we model the node observations as $\mathbf{X}_i =$

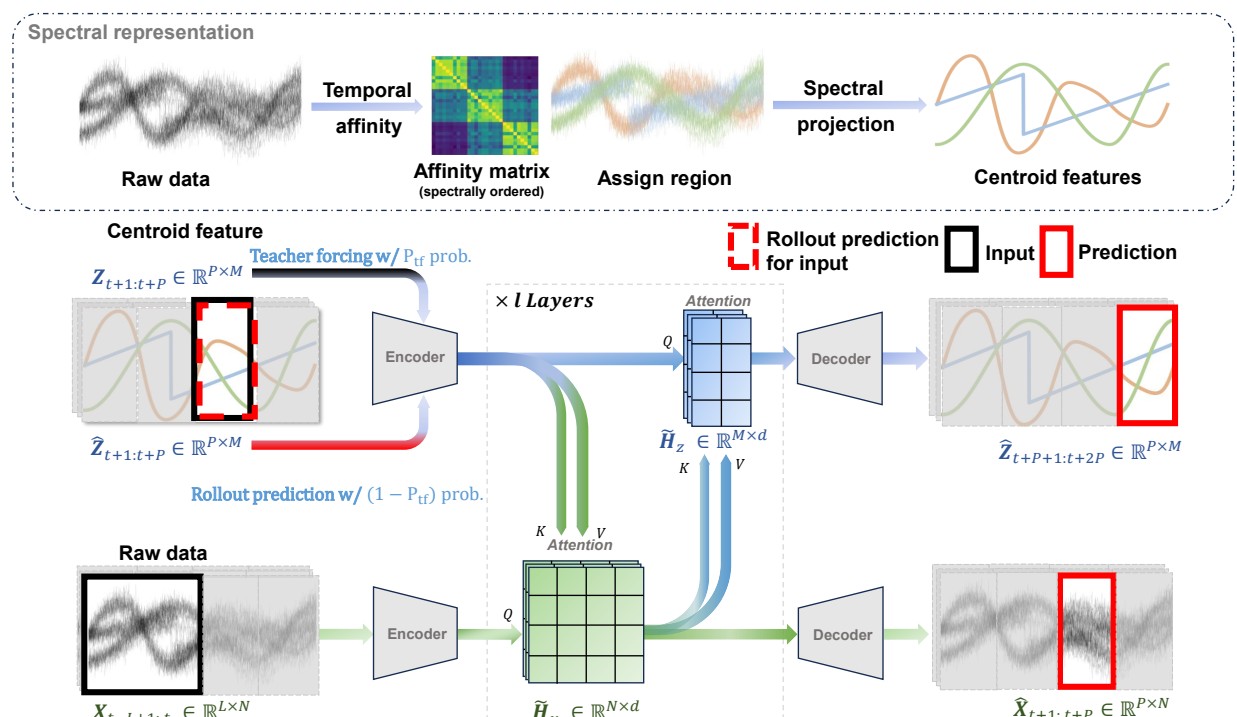

*Figure 1.* Diagram of NEST. (i) *Spectral representation:* Node-level time series are partitioned via spectral clustering to derive representative regional dynamics **Z**, which serve as macroscopic structural anchors for the system. (ii) *Training phase:* We process historical node signals $\mathbf{X}_{t-L+1:t}$ alongside future regional guidance $\mathbf{Z}_{t+1:t+P}$ using decoupled encoders. To bridge the discrepancy between training and inference, we employ a scheduled sampling strategy: ground-truth regional features are provided via teacher-forcing with probability $P_{\text{tf}}$, while predicted rollouts $\hat{\mathbf{Z}}_{t+1:t+P}$ are utilized with probability $1 - P_{\text{tf}}$. Bidirectional information flow between scales is facilitated via cross-attention (MLP is omitted for clarity), which effectively bounds the interaction complexity to the number of clusters $M$ (where $M < N$), ensuring future-oriented guidance while maintaining linear scalability relative to the number of nodes.

$\mathbf{S}_i + \epsilon_i$. Theorem 1 demonstrates that aggregating nodes with similar patterns into latent regions significantly enhances the Signal-to-Noise Ratio (SNR).

**Theorem 1.** *Consider a graph signal where the noise at each node is independent. Let $\mathcal{C}_m$ denote a cluster of nodes with cardinality $|\mathcal{C}_m|$, and let $\mathbf{Z}_m$ be the corresponding cluster center. Then the SNR of the cluster center satisfies*

$$\text{SNR}(\mathbf{Z}_m) \geq [1 + (|\mathcal{C}_m| - 1)\rho_m] \cdot \overline{\text{SNR}}_m, \quad (4)$$

*where $\overline{\text{SNR}}_m$ is the average SNR of the individual signals within cluster $\mathcal{C}_m$, and $\rho_m$ is the average correlation coefficient among the true signals in the cluster:*

$$\rho_m = \frac{1}{|\mathcal{C}_m|(|\mathcal{C}_m| - 1)} \sum_{i \neq j \in \mathcal{C}_m} \text{Corr}(\mathbf{s}_i, \mathbf{s}_j). \quad (5)$$

This theorem provides the theoretical guarantee that our region-level modeling effectively suppresses local noise by maximizing intra-cluster correlation $\rho_m$; detailed proofs and derivations are provided in Appendix A.4.

### 4.2. Nested Spatio-Temporal Forecasting

**Dual-Horizon Encoding.** To capture multi-scale dynamics across time and facilitate information interaction, we employ a projection layer that maps sequenced data $\mathbf{X}$ and $\mathbf{Z}$ into a $d$-dimensional space. At time-step $t$, the model processes two inputs that occupy entirely different temporal horizons: the historical node-level sequence $\mathbf{X}_{t-L+1:t} \in \mathbb{R}^{N \times L \times C}$, covering the past $L$ steps, and the regional guidance $\mathbf{Z}_{t+1:t+P} \in \mathbb{R}^{M \times P \times C}$, providing representative features for the targeted future. To unify these inputs at different scale, we first flatten their temporal and feature axes and project them into the latent space.

$$\mathbf{H}_x^{\text{past}} = \text{Linear}_x(\text{Flatten}(\mathbf{X}_{t-L+1:t})) + \mathbf{TE}_x + \mathbf{SE}_{\text{node}},$$

$$\mathbf{H}_z^{\text{fut}} = \text{Linear}_z(\text{Flatten}(\mathbf{Z}_{t+1:t+P})) + \mathbf{TE}_z + \mathbf{SE}_{\text{region}},$$

where $\mathbf{H}_x^{\text{past}} \in \mathbb{R}^{N \times d}$ and $\mathbf{H}_z^{\text{fut}} \in \mathbb{R}^{M \times d}$ represent the latent tokens for the past node context and future region guidance, respectively. To explicitly encode structural and periodic dynamics, we incorporate spatial embeddings ($\mathbf{SE}_{\text{node}}$ and $\mathbf{SE}_{\text{region}}$) to capture spatial identities, along with temporal embeddings ($\mathbf{TE}_x$ and $\mathbf{TE}_z$), which are learnable Time-of-Day and Day-of-Week features averaged across the horizon.

**Cross-Scale Interaction.** To couple fine-grained historical dynamics with coarse-grained future trends, we introduce a bidirectional cross-attention mechanism. Given a standard attention operator $\text{Attn}(Q, K, V)$, interaction proceeds in two ways.

First, a top-down guidance step allows node-level tokens to query regional tokens:

$$\tilde{\mathbf{H}}_x = \text{Attn}(\mathbf{H}_x^{\text{past}}, \mathbf{H}_z^{\text{fut}}, \mathbf{H}_z^{\text{fut}}), \qquad (6)$$

enabling node representations to incorporate macroscopic evolutionary trends.

Next, a bottom-up refinement step updates regional tokens by querying the enriched node features:

$$\tilde{\mathbf{H}}_z = \text{Attn}(\mathbf{H}_z^{\text{fut}}, \tilde{\mathbf{H}}_x, \tilde{\mathbf{H}}_x). \qquad (7)$$

This step anchors regional guidance in the most recent fine-grained context. Together, these interactions yield representations that are both locally detailed and globally predictive.

**Dual-Head Decoding.** Following cross-scale interaction, we decode both fine- and coarse-grained predictions. A node-level decoding head produces

$$\hat{\mathbf{X}}_{t+1:t+P} = \text{Proj}_x(\tilde{\mathbf{H}}_x),$$

while a regional decoding head predicts the next segment of macro dynamics:

$$\hat{\mathbf{Z}}_{t+P+1:t+2P} = \text{Proj}_z(\tilde{\mathbf{H}}_z).$$

The predicted regional states are recursively used as guidance for subsequent decoding steps, enabling nested auto-regressive generation.

**Teacher forcing and multi-step ahead rollout.** During inference, future regional guidance $\mathbf{Z}_{t+1:t+P}$ is unavailable, whereas training relies on teacher forcing and rollout prediction. To bridge this discrepancy, we first introduce a boundary modeling strategy based on masked guidance reconstruction.

During training, a proportion of regional tokens is replaced with zero masks, and a dedicated boundary decoder is trained to recover the missing guidance:

$$\hat{\mathbf{Z}}_{t+1:t+P} = \text{Proj}_{\text{bd}}\Big(\text{Attn}(\mathbf{H}_z^{\text{zeros}}, \tilde{\mathbf{H}}_x, \tilde{\mathbf{H}}_x)\Big). \qquad (8)$$

, where $\mathbf{H}_z^{\text{zeros}}$ is the hidden state encoded from an all-zero mask, capturing the prior state before cross-scale interaction.

Then, building on this initialized boundary $\hat{\mathbf{Z}}_{t+1:t+P}$, we execute a multi-step rollout. Specifically, we let the rollout prediction for future macro-state $\hat{\mathbf{Z}}_{t+P+1:t+2P}$ be fed back

as the guidance for the next time window. This establishes a coherent auto-regressive loop where evolving macro-trends continuously anchor and regularize the long-term micro-level predictions.

At inference time, guidance is initialized with zero masks, and the reconstructed $\hat{\mathbf{Z}}_{t+1:t+P}$ serves as the structural anchor for node-level prediction. This strategy aligns training and inference behavior and stabilizes early rollout steps.

### 4.3. Uncertainty-Aware guidance and Complexity

**Robustness via Quantile Regression.** To mitigate error accumulation and estimate the uncertainty caused by inaccurate macro-level guidance, we explicitly model regional dynamics as predictive distributions rather than point estimates. We adopt quantile regression (Bassett & Jr., 1978) to estimate multiple conditional quantiles $\{\tau_q\}_{q=1}^{Q}$ of future regional states, thereby capturing epistemic uncertainty in coarse-grained evolution:

$$\hat{\mathbf{Z}}_{t+1:t+P}^{(\tau_q)} = f^{(\tau_q)}\left(\tilde{\mathbf{H}}_z\right), \quad q = 1, \dots, Q. \qquad (9)$$

For inference, we use the median prediction ($\tau = 0.5$) as deterministic guidance for downstream node-level forecasting. This design leverages the inherent stability of macroscopic dynamics while reducing sensitivity to local noise and outliers, leading to more reliable long-horizon predictions. Details of the quantile regression loss are provided in Appendix A.6.

**Training Objective.** The proposed model is trained end-to-end under a multi-task objective that jointly optimizes: (i) fine-grained node-level forecasting, (ii) uncertainty-aware regional forecasting, and (iii) masked guidance reconstruction to handle missing future context. The overall loss is defined as

$$\mathcal{L} = \mathcal{L}_x + \lambda_1 \mathcal{L}_z + \lambda_2 \mathcal{L}_{\text{bd}}, \qquad (10)$$

where $\mathcal{L}_x$ denotes the node-level forecasting loss, $\mathcal{L}_z$ supervises the multi-quantile predictions of regional dynamics, and $\mathcal{L}_{\text{bd}}$ corresponds to the masked guidance reconstruction loss introduced for boundary modeling. The hyperparameters $\lambda_1$ and $\lambda_2$ control the trade-off between fine-grained accuracy, macro-level uncertainty modeling, and robustness to missing guidance. Formal definitions of each loss term are deferred to Appendix A.6.

**Computational Complexity.** We analyze the computational complexity with respect to the number of nodes $N$, regions $M$ ($M < N$), latent dimension $d$, and the number of cross-attention layers $l$. In each layer, the dominant cost arises from cross-attention between node-level and region-level tokens, resulting in a complexity of $\mathcal{O}(NMd)$ per layer and $\mathcal{O}(lNMd)$ per forward pass. In contrast, a standard node-level transformer with full self-attention incurs

$\mathcal{O}(lN^2d)$ complexity. Since $M < N$ in practice, the proposed hierarchical design substantially reduces the quadratic dependence on the number of nodes, achieving near-linear scaling while preserving global contextual modeling.

## 5. Experiments

In this section, we present a comprehensive empirical evaluation on diverse large-scale benchmarks to validate the effectiveness and robustness of our proposed framework.

### 5.1. Experiment Setup

**Dataset**. We select the GLA, GBA, and CA datasets from the LargeST benchmark (Liu et al., 2023b), prioritizing their high node counts to rigorously test our model's scalability and spatial modeling capabilities. Following established settings (Fang et al., 2025), we chronologically split the data into training, validation, and test sets with a 6:2:2 ratio. The forecasting task is framed as using 12 historical time steps as input to predict the subsequent 12 steps. Detailed datasets are presented in Appendix A.1.

**Baselines**. Our proposed method is compared with 11 advanced baselines to demonstrate its superiority. The benchmark models include STID (Shao et al., 2022b), GWNET (Wu et al., 2019), AGCRN (Bai et al., 2020), STGODE (Fang et al., 2021), DGCRN (Li et al., 2023), DSTAGNN (Lan et al., 2022), D2STGNN (Shao et al., 2022c), STWAVE (Fang et al., 2023), RPMIXER (Yeh et al., 2024), BIGST (Han et al., 2024), and the current state-of-the-art PATCHSTG (Fang et al., 2025). Detailed descriptions of these methods are provided in AppendixA.2.

**Evaluation Metrics.** We evaluate forecasting performance using three standard metrics: Mean Absolute Error (MAE), Root Mean Square Error (RMSE), and Mean Absolute Percentage Error (MAPE). Their formal definitions and calculation formulas are provided in Appendix A.3.

**Implementation Details.** Consistent with existing literature, the forecasting task is configured with a look-back window $L = 12$ and a prediction horizon $H = 12$. Comprehensive implementation details, encompassing hyperparameter settings, optimization strategies, and hardware environments, are provided in Appendix A.5.

### 5.2. Performance Comparisons

**Overall Performance.** Table 1 summarizes the comprehensive performance of NEST against all other baselines. Our framework consistently achieves the top performance across all datasets, metrics, and horizons (3, 6, and 12 steps). On average across all three datasets, NEST improves MAE by 4.71%, RMSE by 4.41%, and MAPE by 9.34%, respectively. Notably, the improvements in MAPE exceed 10% on GBA

and CA, demonstrating a superior ability to handle high-dimensional volatility. This success stems from two core mechanisms: semantic regional aggregation and explicit macro-trend regularization. While baseline models often rely on passive historical aggregation, our approach utilizes future regional guidance to mitigate local noise, ensuring fine-grained node predictions remain aligned with broader semantic context.

**Long-Horizon Stability.** To evaluate the temporal stability of our framework, we compare NEST against the strongest baseline, PATCHSTG, on the GLA and CA over an extended 48-step horizon (12 hours). Crucially, these forecasts were generated via auto-regressive rollout, using the model's own predictions step-by-step to align with realistic deployment scenarios where training separate models for every horizon is impractical. As detailed in Table 2, NEST demonstrates superior robustness, consistently outperforming PATCHSTG across all extended horizons. Although auto-regressive inference typically suffers from cumulative error propagation, our framework mitigates this by conditioning nodal forecasts on predicted regional trends. For example, on the GLA dataset, the performance gap in our favor expands from 2.0 MAE at step 16 to 2.4 MAE at step 48. These results confirm that leveraging macro-trends guidance as top-down context effectively stabilizes long-term rollouts and maintains coherence over extended periods.

**Generalization to Non-Traffic Domains.** While our primary evaluation focuses on large-scale traffic networks, we further verify the broad applicability of NEST on diverse non-traffic datasets, including meteorology (KnowAir), energy (UrbanEV), and classical time-series benchmarks (Electricity, Solar-Energy). As detailed in Appendix A.13, NEST achieves consistent and superior performance against recent state-of-the-art models (e.g., MAGE (Ma et al., 2025b), Air-DualODE (Tian et al., 2025), and iTransformer (Liu et al., 2024)) across these diverse domains. These supplementary results confirm that our data-driven semantic clustering and cross-scale attention mechanisms successfully capture general spatio-temporal dynamics without relying on traffic-specific physical priors.

### 5.3. Ablation Study

We analyze the contribution of individual components by grouping experiments into two parts: the interaction mechanism (how regions and nodes communicate) and the semantic partitioning strategy (how regions are constructed).

**Impact of Macro-Micro Interaction.** To test our hierarchical coupling design, We compare NEST against two ablated variants: (i) **w/o CA (no Cross-Attention)**, which completely removes the cross-attention module, severing the link between regional and nodal representations; and

*Table 1.* Performance comparison on GBA, GLA, and CA datasets. **Red** indicates the best results, and **Blue** indicates the second-best results. The row "*Improv.*" denotes the relative improvement of our method over the best baseline. NEST consistently outperforms prior methods across datasets and horizons.

| Dataset | Method | Horizon 3 | | | Horizon 6 | | | Horizon 12 | | | Average | | |
|---|---|---|---|---|---|---|---|---|---|---|---|---|---|
| | | MAE | RMSE | MAPE | MAE | RMSE | MAPE | MAE | RMSE | MAPE | MAE | RMSE | MAPE |
| GBA | GWNET (Wu et al., 2019) | 17.85 | 29.12 | 13.92 | 21.11 | 33.69 | 17.79 | 25.58 | 40.19 | 23.48 | 20.91 | 33.41 | 17.66 |
| | AGCRN (Bai et al., 2020) | 18.31 | 30.24 | 14.27 | 21.27 | 34.72 | 16.89 | 24.85 | 40.18 | 20.80 | 21.01 | 34.25 | 16.90 |
| | STGODE (Fang et al., 2021) | 18.84 | 30.51 | 15.43 | 22.04 | 35.61 | 18.42 | 26.22 | 42.90 | 22.83 | 21.79 | 35.37 | 18.26 |
| | DSTAGNN (Lan et al., 2022) | 19.73 | 31.39 | 15.42 | 24.21 | 37.70 | 20.09 | 30.12 | 46.40 | 28.16 | 23.82 | 37.29 | 20.16 |
| | D2STGNN (Shao et al., 2022c) | 17.54 | 28.94 | 12.12 | 20.92 | 33.92 | 14.89 | 25.48 | 40.99 | 19.83 | 20.71 | 33.65 | 15.04 |
| | DGCRN (Li et al., 2023) | 18.02 | 29.49 | 14.13 | 21.08 | 34.03 | 16.94 | 25.25 | 40.63 | 21.15 | 20.91 | 33.83 | 16.88 |
| | STID (Shao et al., 2022b) | 17.36 | 29.39 | 13.28 | 20.45 | 34.51 | 16.03 | 24.38 | 41.33 | 19.90 | 20.22 | 34.61 | 15.91 |
| | STWAVE (Fang et al., 2023) | 17.95 | 29.42 | 13.01 | 20.99 | 34.01 | 15.62 | 24.96 | 40.31 | 20.08 | 20.81 | 33.77 | 15.76 |
| | RPMIXER (Yeh et al., 2024) | 20.31 | 33.34 | 15.64 | 26.95 | 44.02 | 22.75 | 39.66 | 66.44 | 37.35 | 27.77 | 47.72 | 23.87 |
| | BIGST (Han et al., 2024) | 18.70 | 30.27 | 15.55 | 22.21 | 35.33 | 18.54 | 26.98 | 42.73 | 23.68 | 21.95 | 35.54 | 18.50 |
| | PATCHSTG (Fang et al., 2025) | 16.81 | 28.71 | 12.25 | 19.68 | 33.09 | 14.51 | 23.49 | 39.23 | 18.93 | 19.50 | 33.16 | 14.64 |
| | NEST | **16.05** | **27.53** | **10.66** | **18.90** | **31.77** | **12.86** | **22.63** | **37.71** | **16.13** | **18.73** | **31.85** | **12.90** |
| | *Improv.* | 4.52% | 4.11% | 12.98% | 3.96% | 3.99% | 11.37% | 3.66% | 3.87% | 14.79% | 3.95% | 3.95% | 11.88% |
| GLA | GWNET (Wu et al., 2019) | 17.28 | 27.68 | 10.18 | 21.31 | 33.70 | 13.02 | 26.99 | 42.51 | 17.64 | 21.20 | 33.58 | 13.18 |
| | AGCRN (Bai et al., 2020) | 17.27 | 29.70 | 10.78 | 20.38 | 34.82 | 12.70 | 24.59 | 42.59 | 16.03 | 20.25 | 34.84 | 12.87 |
| | STGODE (Fang et al., 2021) | 18.10 | 30.02 | 11.18 | 21.71 | 36.46 | 13.64 | 26.45 | 45.09 | 17.60 | 21.49 | 36.14 | 13.72 |
| | DSTAGNN (Lan et al., 2022) | 19.49 | 31.08 | 11.50 | 24.27 | 38.43 | 15.24 | 30.92 | 48.52 | 20.45 | 24.13 | 38.15 | 15.07 |
| | STID (Shao et al., 2022b) | 16.54 | 27.73 | 10.00 | 19.98 | 34.23 | 12.38 | 24.29 | 42.50 | 16.02 | 19.76 | 34.56 | 12.41 |
| | STWAVE (Fang et al., 2023) | 17.48 | 28.05 | 10.06 | 21.08 | 33.58 | 12.56 | 25.82 | 41.28 | 16.51 | 20.96 | 33.48 | 12.70 |
| | RPMIXER (Yeh et al., 2024) | 19.94 | 32.54 | 11.53 | 27.10 | 44.87 | 16.58 | 40.13 | 69.11 | 27.93 | 27.87 | 48.96 | 17.66 |
| | BIGST (Han et al., 2024) | 18.38 | 29.40 | 11.68 | 22.22 | 35.53 | 14.48 | 27.98 | 44.74 | 19.65 | 22.08 | 36.00 | 14.57 |
| | PATCHSTG (Fang et al., 2025) | 15.84 | 26.34 | 9.27 | 19.06 | 31.85 | 11.30 | 23.32 | 39.64 | 14.60 | 18.96 | 32.33 | 11.44 |
| | NEST | **15.12** | **25.44** | **8.80** | **17.94** | **30.22** | **10.69** | **21.85** | **36.91** | **13.49** | **17.89** | **30.52** | **10.74** |
| | *Improv.* | 4.55% | 3.42% | 5.07% | 5.89% | 5.13% | 5.40% | 6.30% | 6.90% | 7.60% | 5.65% | 5.60% | 6.14% |
| CA | GWNET (Wu et al., 2019) | 17.14 | 27.81 | 12.62 | 21.68 | 34.16 | 17.14 | 28.58 | 44.13 | 24.24 | 21.72 | 34.20 | 17.40 |
| | STGODE (Fang et al., 2021) | 17.57 | 29.91 | 13.91 | 20.98 | 36.62 | 16.88 | 25.46 | 45.99 | 21.00 | 20.77 | 36.60 | 16.80 |
| | STID (Shao et al., 2022b) | 15.51 | 26.23 | 11.26 | 18.53 | 31.56 | 13.82 | 22.63 | 39.37 | 17.59 | 18.41 | 32.00 | 13.82 |
| | STWAVE (Fang et al., 2023) | 16.77 | 26.98 | 12.20 | 18.97 | 30.69 | 14.40 | 25.36 | 38.77 | 19.01 | 19.69 | 31.58 | 14.58 |
| | RPMIXER (Yeh et al., 2024) | 18.18 | 30.49 | 12.86 | 24.33 | 41.38 | 18.34 | 35.74 | 62.12 | 30.38 | 25.07 | 44.75 | 19.47 |
| | BIGST (Han et al., 2024) | 17.15 | 27.92 | 13.03 | 20.44 | 33.16 | 15.87 | 25.49 | 41.09 | 20.97 | 20.32 | 33.45 | 15.91 |
| | PATCHSTG (Fang et al., 2025) | 14.69 | 24.82 | 10.51 | 17.41 | 29.43 | 12.83 | 21.20 | 36.13 | 16.00 | 17.35 | 29.79 | 12.79 |
| | NEST | **13.98** | **24.17** | **9.37** | **16.59** | **28.30** | **11.24** | **20.31** | **34.44** | **14.18** | **16.54** | **28.55** | **11.28** |
| | *Improv.* | 4.82% | 2.62% | 10.81% | 4.73% | 3.86% | 12.41% | 4.17% | 4.69% | 11.18% | 4.69% | 4.17% | 11.78% |

*Table 2.* Long-horizon **MAE** comparison on GLA and CA datasets. We evaluate performance across increasing prediction steps (hours). Best results are highlighted in **bold**.

| Dataset | Model | Prediction Horizon (Steps / Hours) | | | | |
|---|---|---|---|---|---|---|
| | | **16** (4h) | **20** (5h) | **24** (6h) | **36** (9h) | **48** (12h) |
| GLA | PATCHSTG | 25.63 | 27.08 | 27.92 | 30.42 | 32.43 |
| | NEST | **23.63** | **25.11** | **26.28** | **28.51** | **30.03** |
| CA | PATCHSTG | 24.17 | 25.52 | 26.54 | 28.10 | 29.15 |
| | NEST | **22.05** | **23.46** | **24.56** | **26.60** | **27.92** |

(ii) **w/o FG (no Future Guidance)**, which replaces the predicted future regional signals $\mathbf{Z}_{t+1:t+P}$ with past region observations $\mathbf{Z}_{t-P+1:t}$. As detailed in Table 3, both components prove to be essential for optimal performance. The removal of cross-attention (**w/o CA**) results in the most significant degradation, raising MAE by 1.04 on GBA and 1.11 on GLA. This substantial drop indicates that without the top-down regularization from region-level semantics, the model struggles to handle local noise effectively, degenerating into a purely local predictor. Similarly, the **w/o FG** variant, which relies solely on past regional patterns, fails to capture evolving temporal shifts. This leads to a notable increase

in RMSE (e.g., +1.04 on GBA), suggesting that historical context alone is insufficient for predictive stability. Overall, these results show that explicitly modeling region–node interactions together with future-aware guidance is crucial for robust multi-step forecasting in NEST.

*Table 3.* Ablation results of Macro-Micro Interaction on GBA and GLA. Best results are highlighted in **bold**.

| Variant | GBA | | | GLA | | |
|---|---|---|---|---|---|---|
| | MAE | RMSE | MAPE (%) | MAE | RMSE | MAPE (%) |
| w/o CA | 19.76 | 34.11 | 15.33 | 19.00 | 32.76 | 11.37 |
| w/o FG | 19.64 | 32.89 | 14.88 | 18.85 | 32.04 | 11.03 |
| NEST | **18.73** | **31.85** | **12.89** | **17.89** | **30.52** | **10.74** |

**Effectiveness of Semantic Partitioning.** We evaluate our spatial partitioning strategy against four baselines: (i) **w/ KM (K-Means)**, which clusters nodes based on raw features similarity; (ii) **w/ RN (Random Node)**, which selects $M$ individual nodes as regional representatives; (iii) **w/ RP (Random Partition)**, which randomly assigns nodes to clusters; (iv) **w/ DA (Distance Adjacency)**, which uses static geographic proximity for spectral clustering. As shown in

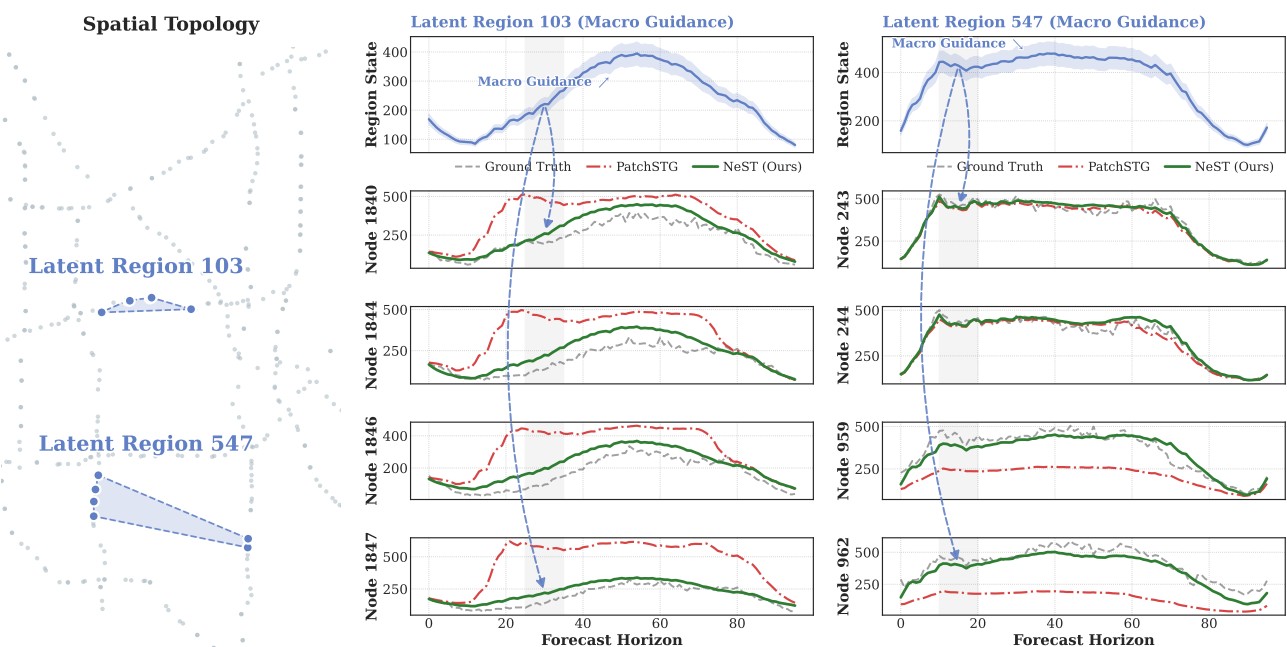

*Figure 2.* **Illustration of the Nested Spatio-Temporal Forecasting mechanism.** The left panel shows the spatial distribution of two learned latent regions. The right panels visualize the two-stage decoding process: (1) Macro Generation, where the model first predicts a stable regional trend (see top row); and (2) Micro Guidance, where this trend explicitly guides the forecasting of downstream nodes (bottom rows, indicated by arrows). As shown in the highlighted intervals, this mechanism enables NEST (red) to maintain robustness against local noise and align better with the Ground Truth (gray dashed) compared to PATCHSTG (green).

Table 4, the quality of regional definitions is pivotal. Arbitrary grouping (**w/ RP**) leads to a 13% MAPE increase on GBA due to the lack of coherent macro patterns, while single-node representation (**w/ RN**) suffers from sensitivity to local noise. Crucially, our semantic approach outperforms both **w/ KM** and **w/ DA**, reducing MAE by 0.21 on GBA and 0.45 on GLA. Unlike **w/ KM**, which ignores graph topology, or **w/ DA**, which is limited by static geometry, our method captures functional spatial dependencies, providing a robust foundation for stable regional modeling.

*Table 4.* Ablation results of Semantic Partitioning Strategy on GBA and GLA. Best results are highlighted in **bold**.

| Variant | GBA | | | GLA | | |
|---|---|---|---|---|---|---|
| | MAE | RMSE | MAPE (%) | MAE | RMSE | MAPE (%) |
| w/ KM | 18.93 | 32.33 | 13.46 | 18.39 | 31.55 | 10.82 |
| w/ RN | 19.44 | 32.96 | 14.42 | 18.46 | 32.42 | 10.85 |
| w/ RP | 19.07 | 32.47 | 14.27 | 18.46 | 31.50 | 11.22 |
| w/ DA | 18.93 | 32.22 | 13.37 | 18.34 | 31.99 | 11.04 |
| NEST | **18.73** | **31.85** | **12.89** | **17.89** | **30.52** | **10.74** |

### 5.4. Computational Efficiency and Runtime Analysis

Table 5 compares the empirical runtime of NEST against the strong baseline PatchSTG on the GBA and GLA datasets, with all tests conducted on a single RTX 4090 GPU using a batch size of 64. NEST substantially accelerates both

the training and inference stages. Specifically, on the GBA dataset, the training time per epoch drops from 185 to 75 seconds (a 59.5% reduction), and the total inference time decreases from 32 to 20 seconds (a 37.5% reduction). Similar efficiency gains are also observed on the larger GLA dataset. While NEST requires an offline preprocessing step for affinity matrix construction and spectral clustering (e.g., 91 seconds on the GBA dataset), this procedure is executed only once. Consequently, this one-time overhead is negligible relative to the overall training process. These results demonstrate that the proposed linear cross-attention mechanism and patch-wise prediction design effectively constrain computational complexity, making the framework highly efficient in practice.

*Table 5.* Empirical runtime comparison on the GBA and GLA datasets (Batch Size = 64, single RTX 4090 GPU). NEST significantly reduces both training and inference costs.

| Dataset | Model | Training Time (s/epoch) | Inference Time (s) |
|---|---|---|---|
| GBA | PatchSTG | 185 | 32 |
| | NEST **(Ours)** | **75** | **20** |
| GLA | PatchSTG | 235 | 42 |
| | NEST **(Ours)** | **137** | **37** |

### 5.5. In-Depth Analysis

**Case Study.** Figure 2 visualizes the nested forecasting mechanism on two representative latent regions. **Spatial**

**Distribution (Left):** The learned regions successfully group nodes with synchronized traffic patterns, regardless of physical distance. For instance, nodes in Region 547 are distributed across disparate road segments yet share a consistent traffic evolution, confirming that our model captures semantic functional similarities beyond mere geographical proximity. **Mechanism Analysis (Right):** The right panels illustrate the coarse-to-fine generation process. At the macro-level (top plots), NEST first extracts distinct regional trends, such as the bell-shaped pattern in Region 103 and the sharp decline in Region 547. Explicitly guided by these macro-signals (indicated by the dashed arrows), our model (red solid line) achieves precise micro-level predictions. As observed in the highlighted intervals, the macro guidance effectively directs individual nodes to adapt to sudden traffic shifts. Consequently, NEST aligns closely with the Ground Truth (grey line), whereas the baseline PATCHSTG (green dashed line) fails to capture these dynamics, exhibiting significant lag or overestimation during trend transitions.

**Sensitivity to Region Count** $M$. We investigate the sensitivity of NEST to the number of latent regions $M$ on both GBA and GLA datasets. The results reveal a consistent U-shaped pattern, achieving optimal performance at $M = 0.2N$. To further validate this, we conduct a fine-grained analysis on the GBA dataset (detailed in Appendix A.12). Specifically, prediction errors decrease as $M$ increases from $0.1N$ to $0.2N$ but rebound when $M$ is further increased to $0.3N$. We attribute this phenomenon to a critical trade-off in granular modeling: a small $M$ ($0.1N$) likely causes over-aggregation, smoothing out distinct local patterns, whereas a large $M$ ($0.3N$) becomes susceptible to structural noise and fails to filter spurious correlations. Consequently, $M = 0.2N$ strikes the optimal balance, effectively abstracting stable macro-trends while preserving necessary micro-dynamics.

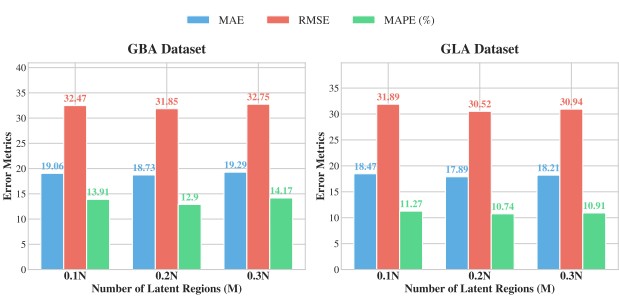

*Figure 3.* Forecasting performance with varying numbers of latent regions ($M$) on GBA and GLA.

## 6. Conclusion

In this paper, we proposed NEST, a novel macro-to-micro framework designed to tackle local noise sensitivity and cascading error accumulation in high-dimensional spatio-temporal forecasting. By leveraging data-driven semantic

clustering without relying on physical priors, NEST extracts stable regional representations that act as a natural low-pass filter against local volatility. Crucially, explicitly predicted future macro-trends serve as top-down guidance to regularize micro-level dynamics, inherently anchoring the autoregressive rollout and mitigating long-horizon error propagation. Extensive experiments across diverse domains, including traffic, meteorology, energy, and classical time series, demonstrate that NEST achieves state-of-the-art accuracy and exceptional long-term stability. Ultimately, this work provides a highly generalizable paradigm for multi-scale sequence modeling in complex, real-world environments.

## Impact Statement

This work advances the field of Machine Learning by proposing a robust framework for spatio-temporal time-series forecasting. The proposed method has potential applications in smart city operations, energy management, and environmental monitoring, which can lead to more efficient and sustainable infrastructure. We are not aware of any immediate negative societal consequences or specific ethical issues associated with this methodological research, provided that standard data privacy and fairness protocols are followed during deployment.

While NEST demonstrates strong performance, we acknowledge several limitations. First, constructing the feature-driven affinity matrix introduces a preprocessing overhead that scales quadratically ($\mathcal{O}(N^2)$). Second, our global clustering assumes time-invariant spatial correlations, limiting the model's adaptability to sudden topological shifts or transient events. Third, training via future macro-trend reconstruction (teacher forcing) introduces an exposure bias, creating a training-inference gap during autoregressive generation. Finally, although autoregressive rollout stabilizes long-horizon errors, its sequential nature is slower than direct multi-step models, introducing inference latency that may pose challenges for strict real-time applications at massive scales. Addressing these constraints via dynamic regionalization and scheduled sampling remains a promising avenue for future work.

## Acknowledgements

This work was supported by the Pujiang Talent Program (No. 2025PJA729), National Natural Science Foundation of China (Grant Nos. 82394432, 92249302, and 62276127), the Shanghai Municipal Science and Technology Major Project (Grant No. 2023SHZDZX02), the Brain Science and Brain-like Intelligence Technology - National Science and Technology Major Project (Grant No.2021ZD0201300), the Fundamental and Interdisciplinary Disciplines Break-

through Plan of the Ministry of Education of China (Grant No. JYB2025XDXM118), and the "111" Center (Grant No. B26023).

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

# A. Appendix

## A.1. Dataset.

Our experiments utilize the GLA, GBA, and CA subsets from the LargeST benchmark (Liu et al., 2023b), which comprise traffic data collected via the CalTrans PeMS sensor network. We focus on records from 2019, aggregated into 15-minute intervals (96 daily records). As summarized in Table 6, these datasets provide a large-scale evaluation environment, covering diverse regional scales with node counts ranging from 2,352 to 8,600 and totaling over 300 million observations.

*Table 6.* Dataset statistics.

| Datasets | #Nodes | #Samples | #TimeSlices | Timespan |
|----------|--------|----------|-------------|----------|
| GBA | 2352 | 82M | 35040 | 01/01/2019-12/31/2019 |
| GLA | 3834 | 134M | 35040 | 01/01/2019-12/31/2019 |
| CA | 8600 | 301M | 35040 | 01/01/2019-12/31/2019 |

## A.2. Baselines

We compare the proposed approach with the following advanced baselines:

- STID: It proposes an efficient MLP baseline that solves sample indistinguishability in forecasting by incorporating spatial and temporal identity information.

- GWNET: It captures hidden spatial dependencies using a learned adaptive matrix and models long-range temporal trends via stacked dilated 1D convolutions.

- AGCRN: It employs node-adaptive parameter learning and automatic graph generation to capture fine-grained spatial-temporal correlations without requiring pre-defined graphs.

- STGODE: It captures synchronous spatial-temporal dynamics using tensor-based ordinary differential equations and semantic adjacency matrices.

- DGCRN: It leverages hyper-networks to generate dynamic filter parameters and time-varying graphs from node attributes, integrating them with static structures while optimizing efficiency by limiting decoder iterations.

- DSTAGNN: It replaces pre-defined graphs with a data-driven dynamic graph, using multi-head attention and multi-scale gated convolutions to capture complex spatial-temporal dependencies.

- D2STGNN: It uses an estimation gate and residual decomposition to decouple and independently model diffusion and inherent traffic signals.

- STWAVE: It decouples traffic into trends and events using a dual-channel framework with wavelet-based positional encoding and a query sampling strategy.

- RPMIXER: It models temporal dynamics via MLPs and spatial relationships through the integration of random projection layers.

- BIGST: It is a spatial-temporal graph neural networks characterized by linear complexity, which allows for the efficient exploitation of long-range dependencies in large-scale traffic forecasting task.

- PATCHSTG: It groups nodes via KDTree-based irregular spatial patching and utilizes depth and breadth attention to model local and global dependencies.

## A.3. Evaluation Metrics

To comprehensively evaluate the model performance, and in line with previous works (Fang et al., 2025), we utilize three standard metrics: Mean Absolute Error (MAE), Root Mean Squared Error (RMSE), and Mean Absolute Percentage Error (MAPE). These metrics provide a multi-faceted assessment of prediction accuracy, capturing both average deviations and sensitivity to extreme values. Let $y_i$ denote the ground truth, $\hat{y}_i$ represent the predicted value, and $n$ be the total number of samples; the specific definitions and characteristics of these metrics are as follows:

- **Mean Absolute Error (MAE):** This metric calculates the average of the absolute differences between the predicted and actual values, providing a linear penalty for all errors. Because it does not square the deviations, MAE is more robust to outliers and represents the basic magnitude of the forecasting error.

$$\text{MAE} = \frac{1}{n} \sum_{i=1}^{n} |y_i - \hat{y}_i| \tag{11}$$

- **Root Mean Squared Error (RMSE):** By squaring the errors before averaging, RMSE assigns a significantly higher weight to large deviations. This makes it a sensitive indicator of the model's stability and its ability to avoid large-scale prediction failures in critical traffic scenarios.

$$\text{RMSE} = \sqrt{\frac{1}{n} \sum_{i=1}^{n} (y_i - \hat{y}_i)^2} \tag{12}$$

- **Mean Absolute Percentage Error (MAPE):** This metric expresses the error as a percentage of the ground truth, offering a scale-independent measure. It is particularly valuable in traffic forecasting for comparing performance across different road segments or time periods with varying traffic volumes.

$$\text{MAPE} = \frac{1}{n} \sum_{i=1}^{n} \left| \frac{y_i - \hat{y}_i}{y_i} \right| \times 100\% \tag{13}$$

### A.4. Proof of Theorem 1

**Theorem**: Let $\mathcal{C}_m$ be a cluster of size $|\mathcal{C}_m|$ with center $\mathbf{Z}_m$. The SNR of $\mathbf{Z}_m$ satisfies:

$$\text{SNR}(\mathbf{Z}_m) \geq [1 + (|\mathcal{C}_m| - 1)\rho_m] \cdot \overline{\text{SNR}}_m, \tag{14}$$

where $\overline{\text{SNR}}_m$ is the average individual SNR, and $\rho_m$ is the average pairwise correlation of true signals $\mathbf{S}$ within the cluster.

*Proof.* We consider a single region $\mathcal{C}_m$. For any node $i \in \mathcal{C}_m$, its signal can be decomposed as $\mathbf{X}_i = \mathbf{S}_i + \boldsymbol{\epsilon}_i$, where $\mathbf{S}_i$ is the deterministic signal component and $\boldsymbol{\epsilon}_i \sim \mathcal{N}(0, \sigma^2 \mathbf{I}_C)$ is i.i.d. noise. Then, we define $\bar{\mathbf{S}}_m = \frac{1}{n_k} \sum_{i \in \mathcal{C}_m} \mathbf{S}_i$.

The cluster center is $\mathbf{Z}_m = \frac{1}{|\mathcal{C}_m|} \sum_{i \in \mathcal{C}_m} \mathbf{X}_i = \bar{\mathbf{S}}_m + \frac{1}{|\mathcal{C}_m|} \sum_{i \in \mathcal{C}_m} \boldsymbol{\epsilon}_i$. The power of the signal component in $\mathbf{Z}_m$ is $\|\bar{\mathbf{S}}_m\|^2$. The noise component $\frac{1}{|\mathcal{C}_m|} \sum_i \boldsymbol{\epsilon}_i$ has covariance $\frac{\sigma^2}{|\mathcal{C}_m|} \mathbf{I}_C$. Thus, the SNR of $\mathbf{Z}_m$ is

$$\text{SNR}(\mathbf{Z}_m) = \frac{\|\bar{\mathbf{s}}_m\|^2}{\sigma^2 / |\mathcal{C}_m|} = |\mathcal{C}_m| \cdot \frac{\|\bar{\mathbf{s}}_m\|^2}{\sigma^2}. \tag{15}$$

For an individual node $i \in \mathcal{C}_m$, its SNR is $\text{SNR}(\mathbf{x}_i) = \|\mathbf{S}_i\|^2 / \sigma^2$. The average SNR within the cluster is

$$\overline{\text{SNR}}_m = \frac{1}{|\mathcal{C}_m|} \sum_{i \in \mathcal{C}_m} \frac{\|\mathbf{S}_i\|^2}{\sigma^2}. \tag{16}$$

By the Cauchy–Schwarz inequality,

$$\|\bar{\mathbf{S}}_m\|^2 = \frac{1}{|\mathcal{C}_m|^2} \left\| \sum_i \mathbf{S}_i \right\|^2 \geq \frac{1}{|\mathcal{C}_m|^2} \left( \sum_i \|\mathbf{S}_i\| \right)^2. \tag{17}$$

Furthermore, recall the average correlation coefficient $\rho_m$ among the $\mathbf{s}_i$ is defined by:

$$\rho_m \triangleq \frac{1}{|\mathcal{C}_m|(|\mathcal{C}_m| - 1)} \sum_{i \neq j \in \mathcal{C}_m} \text{Corr}(\mathbf{S}_i, \mathbf{S}_j)$$

$$= \frac{1}{|\mathcal{C}_m|(|\mathcal{C}_m| - 1)} \sum_{i \neq j \in \mathcal{C}_m} \frac{\mathbf{S}_i^\top \mathbf{S}_j}{\|\mathbf{S}_i\| \|\mathbf{S}_j\|}. \tag{18}$$

We can obtain the final format of $\|\bar{\mathbf{s}}_m\|^2$:

$$\left\|\bar{\mathbf{S}}_m\right\|^2 \geq \frac{1 + (|\mathcal{C}_m| - 1)\rho_m}{|\mathcal{C}_m|} \cdot \frac{1}{|\mathcal{C}_m|} \sum_i \|\mathbf{S}_i\|^2. \tag{19}$$

Substituting this into equation 15 and using equation 16 yields

$$\mathrm{SNR}(\mathbf{Z}_m) \geq [1 + (|\mathcal{C}_m| - 1)\rho_m] \cdot \overline{\mathrm{SNR}}_m. \tag{20}$$

$\square$

The inequality for the global representation $\mathbf{Z}$ follows by noting that $\mathrm{SNR}(\mathbf{Z})$ is essentially a weighted average of the $\mathrm{SNR}(\mathbf{Z}_m)$, and that the maximum possible enhancement is constrained by the region with the smallest intra-cluster correlation $\rho_m$ and size $|\mathcal{C}_m|$. The factor $N/K$ arises from the dimensionality reduction from $N$ nodes to $M$ regions.

This theorem theoretically validates the denoising mechanism of our framework: the SNR gain scales with both cluster size $|\mathcal{C}_m|$ and intra-cluster correlation $\rho_m$. Since our spectral clustering naturally groups nodes with high signal synchronization (maximizing $\rho_m$), the resulting region-level representations $\mathbf{Z}$ effectively suppress independent local noise while preserving the structural signal fidelity.

### A.5. Implementation Details

All experiments were implemented using the PyTorch framework and conducted on a single NVIDIA A100 80GB GPU. We optimize the model using the AdamW (Loshchilov & Hutter, 2019) optimizer with a fixed learning rate of $3 \times 10^{-4}$. To prevent overfitting, we employed an early stopping strategy, terminating the training if the validation loss did not decrease for 30 consecutive epochs. Regarding the model hyperparameters, the dimension of input embeddings was set to 128, while the latent dimension for the attention mechanism was fixed at 256. The number of latent regions $K$ was adaptively set $0.2N$, where $N$ denotes the number of nodes in the GBA, GLA and CA respectively. The single-step prediction horizon $P$ was set to $\{4, 6, 4\}$ for GBA, GLA and CA. Accordingly, the loss balancing coefficients for regional prediction ($\lambda_1$) and boundary predictions ($\lambda_2$) were configured as (0.1, 0.2), (0.3, 0.3), and (0.1, 0.2), respectively.

### A.6. Loss Function Definitions

**Node-level forecasting loss.** For node-level prediction, we adopt the Huber loss to balance robustness to outliers and sensitivity to small errors. Given the prediction error $e = \hat{x} - x$, the Huber loss is defined as

$$\ell_{\mathrm{Huber}}(e) = \begin{cases} \frac{1}{2}e^2, & |e| \leq \delta, \\ \delta(|e| - \frac{1}{2}\delta), & \text{otherwise,} \end{cases} \tag{21}$$

where $\delta$ is a predefined threshold. The node-level loss $\mathcal{L}_x$ is computed by averaging the Huber loss over all nodes and prediction horizons.

**Quantile-based regional forecasting loss.** To model predictive uncertainty at the regional level, we predict a set of quantiles $\{\tau_q\}_{q=1}^Q$ for each region and optimize the corresponding pinball loss. For a given quantile $\tau \in (0, 1)$ and prediction error $e = \hat{z}_\tau - z$, the pinball loss is defined as

$$\rho_\tau(e) = \max(\tau e, (\tau - 1)e). \tag{22}$$

The regional forecasting loss $\mathcal{L}_z$ is obtained by averaging the pinball loss across all regions, quantiles, and prediction horizons.

**Masked guidance reconstruction loss.** The masked guidance reconstruction loss $\mathcal{L}_{\mathrm{bd}}$ applies the same quantile-based pinball loss to the outputs of the guidance decoder. During training, regional inputs are randomly masked, and the decoder is supervised to reconstruct the corresponding future regional states, encouraging robustness to missing or noisy guidance signals.

## A.7. Period-aligned Chunk Partition for Affinity Construction

To construct the feature-driven affinity matrix $\mathbf{A}$ from raw temporal observations, we partition the training sequence into $\tilde{T}$ non-overlapping temporal chunks. The key motivation is to align the chunking scheme with the intrinsic periodicity of the data, so that $\mathbf{A}$ captures similarity in *long-term temporal evolution* rather than short-term fluctuations.

**Empirical periodicity analysis.**   For each dataset, we conduct a preliminary periodicity analysis on the training split (e.g., via autocorrelation / spectral inspection) and observe a clear dominant period of approximately $P = 100$ time steps across all three datasets. Based on this observation, we set the number of chunks to

$$\tilde{T} = 100, \tag{23}$$

and partition the training sequence into 100 consecutive, non-overlapping chunks. For each node, we compute an averaged representation within each chunk, and define pairwise affinities by aggregating the chunk-wise distances, as described in Eq. (1).

**Rationale.**   This period-aligned chunking ensures that each chunk corresponds to a consistent seasonal phase of the underlying temporal process, thereby encouraging $\mathbf{A}$ to encode node-wise similarity in coarse-grained periodic dynamics. As a result, the constructed affinity matrix is less sensitive to transient noise and short-term irregular variations.

## A.8. Effect of Cross-Scale Interaction Depth

We study the sensitivity of our model to the depth of the Cross-Scale Interaction module. Specifically, we vary the number of Cross-Transformer layers in $\{4, 5, 6, 7\}$ while keeping all other hyperparameters unchanged. We report the averaged multi-step forecasting performance over all horizons (Steps 1–12) in terms of MAE, RMSE and MAPE.

*Table 7.* Hyperparameter study on the number of Cross-Transformer layers. Results are averaged over all prediction horizons (Steps 1–12).

| Layers | GBA | | | GLA | | |
|---|---|---|---|---|---|---|
| | MAE | RMSE | MAPE | MAE | RMSE | MAPE |
| 4 | **18.73** | **31.85** | **12.90** | 18.42 | 31.67 | 11.01 |
| 5 | 19.09 | 32.77 | 13.53 | 18.44 | 31.89 | 10.94 |
| 6 | 19.06 | 32.57 | 13.58 | **17.89** | **30.52** | **10.74** |
| 7 | 19.12 | 32.45 | 13.79 | 18.21 | 31.24 | 10.76 |

As shown in Table 7, the performance is generally stable across different Cross-Transformer depths on both datasets. On GBA, the best results are achieved with 4 layers, obtaining the lowest MAE, RMSE, and MAPE, while deeper configurations do not bring further gains and may slightly degrade performance. In contrast, on GLA, increasing the depth to a moderate level is beneficial: 6 layers yield the best MAE, RMSE, and MAPE, whereas further increasing the depth to 7 layers leads to a minor performance drop. Overall, the differences across depths remain small (within approximately 2% MAE variation between the best and worst settings on both datasets), indicating that our model is robust to the choice of Cross-Transformer depth.

## A.9. Effectiveness of Quantile Forecasting

To assess the reliability of our probabilistic forecasting module, we report quantile predictions at levels $\{0.1, 0.5, 0.9\}$, corresponding to the lower bound, median, and upper bound, respectively. As shown in Fig. 4, the quantile forecasts exhibit desirable behavior across different noise regimes. In the high-noise setting, the predicted quantile band provides a calibrated uncertainty estimate, where the interval between the 0.1 and 0.9 quantiles expands to reflect increased stochasticity in the observations. Importantly, this uncertainty band captures the majority of the ground-truth trajectory, indicating that the model can produce informative prediction distributions rather than over-confident point estimates. In contrast, in the low-noise setting, the three quantiles become tightly concentrated and nearly overlap with each other as well as with the ground truth, suggesting that the model correctly reduces predictive uncertainty when the signal is stable. Overall, these results demonstrate that our quantile prediction is both adaptive and well-calibrated, yielding accurate distributional forecasts under varying noise conditions.

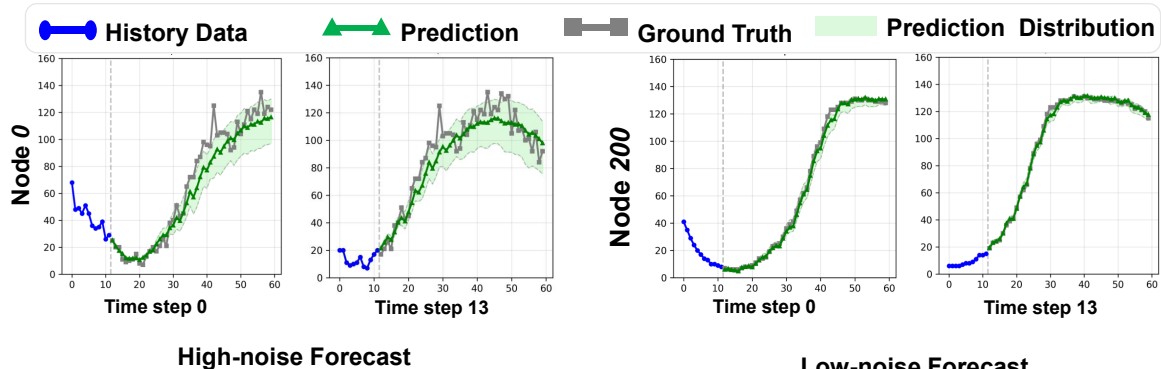

*Figure 4.* **Quantile forecasting under different noise regimes.** We visualize the quantile prediction results using three quantile levels $\{0.1, 0.5, 0.9\}$, where the shaded region denotes the predictive distribution spanned by the lower and upper quantiles. In the high-noise case (left), the quantile band widens to reflect increased uncertainty and covers most of the ground-truth trajectory, indicating well-calibrated uncertainty estimation. In the low-noise case (right), the quantiles collapse into a narrow band and closely match the ground truth, suggesting that the model appropriately reduces uncertainty when the signal is stable.

### A.10. Algorithmic Framework

The overall algorithmic procedure of our framework consists of two main stages: latent structure initialization and model optimization. First, to construct the hierarchical structure from node signals, we employ **Graph Spectral Clustering** (Ng et al., 2001) as detailed in Algorithm 1. This process groups nodes with synchronized patterns into latent regions. Subsequently, the training workflow of NEST, presented in Algorithm 2, integrates a boundary modeling task with a scheduled sampling strategy (teacher forcing decay) to bridge the training-inference gap and jointly optimize the coarse-to-fine forecasting objectives.

---

**Algorithm 1** Graph Spectral Clustering

---

**Input:** Node signals $\mathbf{X} \in \mathbb{R}^{N \times T}$, adjacency $\mathbf{A} \in \mathbb{R}^{N \times N}$, clusters $M$, kernel width $\sigma$, K-Means iterations $K$, initializations $n_{\text{init}}$.

**Output:** Assignment matrix $\mathbf{S} \in \{0, 1\}^{N \times M}$, prototypes $\mathbf{C} \in \mathbb{R}^{M \times T}$.

1: **Preprocess:** $\mathbf{A}_{sym} \leftarrow (\mathbf{A} + \mathbf{A}^{\top})/2$; preprocess $\mathbf{X}$ if needed.

2: **Construct similarity (Gaussian):** $d_{ij} = \|x_i - x_j\|^2$, $\mathbf{W}_{ij} = \exp\left(-\frac{d_{ij}}{2\sigma^2}\right)$, and $\mathbf{W}_{ii} = 0$.

3: Degree $\mathbf{D}_{ii} = \sum_j \mathbf{W}_{ij}$; normalized Laplacian $\mathbf{L}_{sym} = \mathbf{I} - \mathbf{D}^{-1/2}\mathbf{W}\mathbf{D}^{-1/2}$.

4: Compute $M$ smallest eigenvectors $\mathbf{U}_M$ of $\mathbf{L}_{sym}$; row-normalize $\mathbf{U} \leftarrow \text{RowNorm}(\mathbf{U}_M)$.

5: Run K-Means on rows of $\mathbf{U}$ with $n_{\text{init}}$ restarts and $K$ iterations; obtain one-hot assignment $\mathbf{S}$.

6: Prototypes: $\mathbf{C} = (\mathbf{S}^{\top}\mathbf{S})^{-1}\mathbf{S}^{\top}\mathbf{X}$.

7: **return** $\mathbf{S}, \mathbf{C}$.

---

### A.11. Prediction Comparison

To qualitatively evaluate the forecasting performance, we visualize the multi-step prediction results of NEST against the state-of-the-art baseline, PATCHSTG, on representative nodes from the GLA dataset. Figure 5 illustrates the forecasted traffic flows alongside the generated macro-level trends. As observed, the baseline method exhibits significant limitations in capturing rapid trend shifts. For Node 3195 (Left), PATCHSTG fails to anticipate the sharp downward trend, predicting a continuous high-traffic volume (blue dashed line) while the ground truth (grey line) drops significantly. This indicates a lack of awareness of the broader contextual evolution. Conversely, for Node 2264 (Right), PATCHSTG underestimates the traffic demand, predicting an immediate decline to near-zero values, whereas the actual traffic remains high before eventually dropping. In contrast, NEST demonstrates superior robustness. By explicitly conditioning the micro-level generation on the predicted macro-trends (Top Row), our model successfully rectifies these deviations. For Node 3195, the macro-module forecasts a clear downward signal, guiding the node predictor to accurately track the flow reduction. Similarly, for Node 2264, the stable high-traffic macro-signal prevents the model from collapsing to zero early. These comparisons highlight that our hierarchical coupling mechanism effectively mitigates forecasting errors caused by local noise and ensures alignment

**Algorithm 2** NEST Training Algorithm

**Input:** Dataset $\mathcal{D}$, Max Epochs $E$, Decay $\gamma$, Min prob $r$.

1:  Initialize parameters $\theta$, set sampling prob $p_{tf} \leftarrow 1.0$.
2:  **for** epoch $e = 1$ to $E$ **do**
3:      $p_{tf} \leftarrow \max(r, \text{Decay}(p_{tf}, e, \gamma))$
4:      **for** batch $(u_t, z_{t+1}, u_{t+1}, z_{t+2})$ in $\mathcal{D}$ **do**
5:          **1. Boundary Gen:**
6:              $\hat{z}_{t+1} = f_\theta(u_t, \mathbf{0})$
7:          **2. Sampling (Scheduled Sampling):**
8:              Draw mask $m \sim \text{Bernoulli}(p_{tf})$
9:              $\tilde{z}_{t+1} = m \cdot z_{t+1} + (1 - m) \cdot \text{StopGrad}(\hat{z}_{t+1})$
10:         **3. Prediction:**
11:             $\hat{u}_{t+1}, \hat{z}_{t+2} = f_\theta(u_t, \tilde{z}_{t+1})$
12:         **4. Optimization:**
13:             $\mathcal{L}_{task} = \mathcal{L}_u(\hat{u}_{t+1}, u_{t+1}) + \mathcal{L}_{z2}(\hat{z}_{t+2}, z_{t+2}) + \lambda \cdot \mathcal{L}_{z1}(\hat{z}_{t+1}, z_{t+1})$
14:             Update $\theta \leftarrow \text{Optimizer}(\theta, \nabla \mathcal{L}_{task})$
15:     **end for**
16:  **end for**

with the true temporal dynamics.

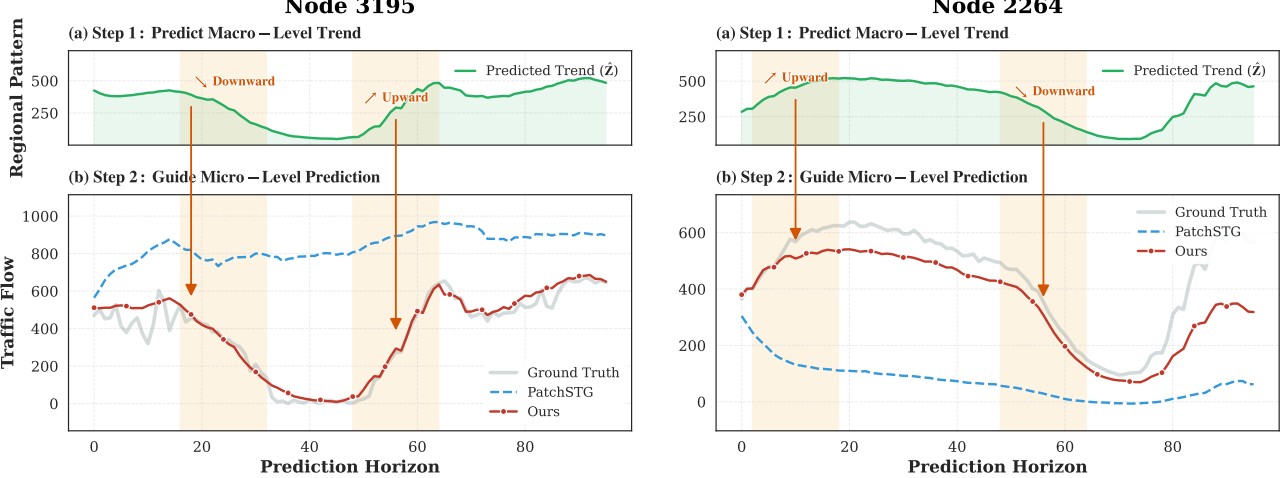

*Figure 5.* **Qualitative comparison of multi-step forecasting results on GBA dataset.** We visualize the predictions for two distinct nodes: Node 3195 (Left) and Node 2264 (Right). **(a) Top Panels:** The macro-level regional trends predicted by our auxiliary module, which serve as guidance. **(b) Bottom Panels:** The comparison of fine-grained node predictions. The baseline PATCHSTG (blue dashed line) struggles to capture sharp transitions, leading to significant overestimation (Node 3195) or underestimation (Node 2264). By incorporating the macro-guidance, NEST (red solid line) effectively corrects these errors, yielding predictions that closely match the **Ground Truth** (grey solid line) across the entire horizon.

## A.12. Fine-grained Sensitivity Analysis of Region Number

*Table 8.* Fine-grained forecasting performance (MAE) with varying numbers of latent regions ($M$) on the GBA dataset.

| Dataset | Number of Regions ($M$) | | | | | | |
|---|---|---|---|---|---|---|---|
| | $0.10N$ | $0.15N$ | $0.18N$ | $0.20N$ | $0.22N$ | $0.25N$ | $0.30N$ |
| GBA | 19.06 | 19.03 | 18.89 | **18.73** | 18.96 | 19.14 | 19.29 |

To comprehensively evaluate the robustness of our spatial granularity design, Table 8 provides the detailed, fine-grained

forecasting performance of NEST with varying numbers of latent regions ($M$) specifically on the GBA dataset.

### A.13. Generalization to Non-Traffic Domains

To further verify the applicability and robustness of NEST beyond traffic forecasting, we evaluate it on diverse non-traffic spatio-temporal datasets as well as classical long-term time series benchmarks.

**Non-Traffic Spatio-Temporal Forecasting.** We first evaluate NEST on two representative non-traffic datasets: **KnowAir** from the meteorology domain and **UrbanEV** from the energy domain. We compare our method with recent state-of-the-art spatio-temporal baselines, including Air-DualODE (Tian et al., 2025), PatchSTG, and MAGE (Ma et al., 2025b). As shown in Table 9, NEST consistently achieves the best performance on both datasets. These results indicate that our data-driven semantic clustering and cross-scale attention mechanisms can effectively capture underlying spatio-temporal dynamics without relying on traffic-specific physical priors.

*Table 9.* Performance comparison (MAE / RMSE) on non-traffic spatio-temporal forecasting datasets.

| Model | KnowAir (Meteorology) | UrbanEV (Energy) |
|---|---|---|
| Air-DualODE | 18.64 / 29.37 | - |
| PatchSTG | 16.08 / 24.70 | 5.16 / 11.53 |
| MAGE | 15.36 / 23.42 | 4.95 / 11.00 |
| NEST (**Ours**) | **14.87 / 22.89** | **4.37 / 10.26** |

**Classical Long-Term Time Series Forecasting.** We further evaluate NEST on two standard long-term forecasting benchmarks, **Electricity** and **Solar-Energy**, to assess its general time series modeling capability. Following the common setup, both the input length and prediction horizon are set to 96 (96 → 96). We compare NEST with strong temporal transformer-based baselines, including PatchTST (Nie et al., 2023), iTransformer (Liu et al., 2024), and TimeMixer (Wang et al., 2024b). As reported in Table 10, NEST achieves competitive or superior performance on both datasets, demonstrating that the proposed macro-to-micro guidance paradigm is effective not only for irregular spatial graphs, but also for general multivariate time series forecasting.

*Table 10.* Performance comparison (MSE / MAE) on classical long-term time series forecasting benchmarks (96 → 96).

| Model | Electricity | Solar-Energy |
|---|---|---|
| PatchTST | 0.190 / 0.296 | 0.265 / 0.323 |
| iTransformer | 0.148 / 0.240 | 0.203 / 0.237 |
| TimeMixer | 0.153 / 0.247 | **0.189** / 0.259 |
| NEST (**Ours**) | **0.141 / 0.235** | 0.201 / **0.211** |

### A.14. Visualization of Centroid Features and Noise Filtering

To intuitively illustrate how the proposed macro-to-micro paradigm handles local anomalies, Figure 6 compares raw micro-node sequences with their corresponding macro centroid features. While individual node trajectories often contain high-frequency noise and sharp local fluctuations, the macro centroids effectively preserve the smoother regional trends. This observation suggests that semantic clustering serves as a natural low-pass filter. By providing stable regional anchors, the cross-scale attention mechanism can more reliably correct noisy micro-level predictions. In this way, macro-level consistency helps shield node-level forecasting from localized anomalies and transient structural noise.

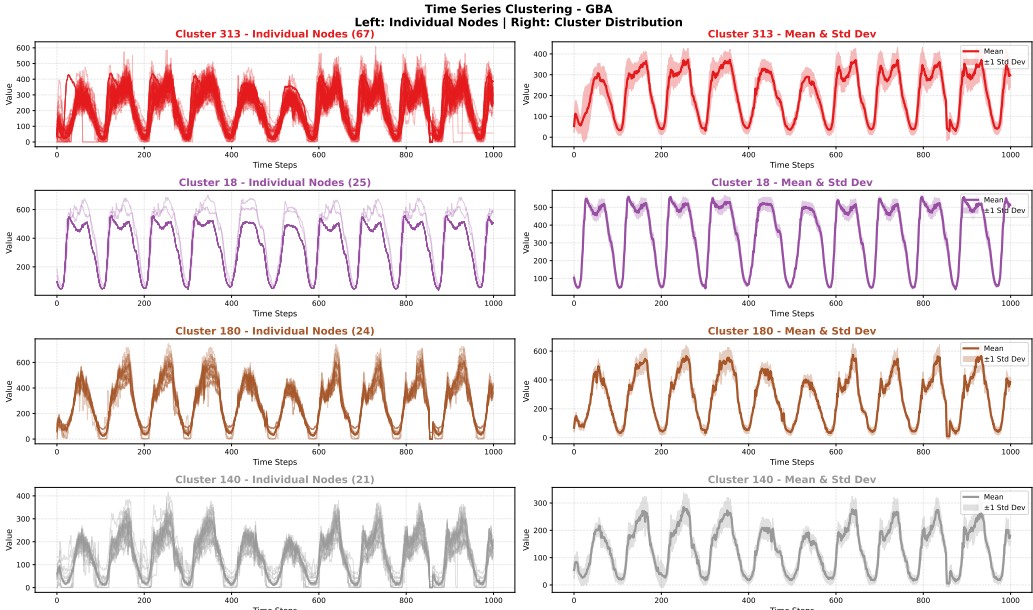

*Figure 6.* Visual comparison between raw node sequences (exhibiting high-frequency noise) and their corresponding macro centroid features (preserving smooth trends).

