# OpenReview forum: "Nested Spatio-Temporal Time Series Forecasting"
_ICML.cc/2026/Conference — ICML 2026 regular_

### Official Review · Reviewer_fSep · 2026-03-08

**Soundness:** 3
**Presentation:** 2
**Significance:** 3
**Originality:** 3
**Overall Recommendation:** 4
**Confidence:** 5

**Summary:**

This paper studies large-scale spatio-temporal forecasting and proposes NeST, a hierarchical coarse-to-fine framework that first constructs region-level centroid series from node signals via a data-driven affinity matrix and spectral clustering, then uses predicted future macro-level regional trends to guide fine-grained node forecasting through cross-attention. The method combines region construction, dual-horizon encoding, cross-scale interaction, dual-head decoding, scheduled sampling, and quantile-based regional prediction. Experiments on three LargeST traffic datasets, GBA, GLA, and CA, show consistent gains over a strong set of baselines, including PatchSTG, and the paper further reports long-horizon and ablation results.

**Compliance With Llm Reviewing Policy:**

Affirmed.

**Final Justification:**

The paper has a meaningful and interesting core idea, and after reading the authors’ response, I now recognize the paper’s experimental strength more clearly. In particular, the response addressed my main concerns about the technical contribution and clarified the aspects that I previously found less convincing.

I still think the paper would benefit from improvements in clarity and presentation, especially in the description of the full pipeline. However, I believe the proposed revisions would significantly improve the quality of the manuscript. Taking both the paper and the authors’ response into account, I now judge the work to have sufficient soundness, originality, and significance to merit a higher score. For this reason, I am increasing my rating from 3 to 4.

**Key Questions For Authors:**

1. **Can the authors reconcile Figure 1, Section 4.2, and Algorithm 2 into one unambiguous training and inference procedure?**
   In particular, does $Z_{t+1:t+P}$ directly participate in node prediction during training, and if so, when is it ground truth, when is it reconstructed by the boundary decoder, and when is it replaced by rollout predictions? A clear answer here would directly affect my soundness and presentation assessment.
2. **Please define the missing symbols and resolve the apparent circular dependency in Eq. (8).**
   What is $H_z^{zeros}$ exactly? How is TE implemented? If $\tilde H_x$ depends on future regional guidance through Eq. (6), how can Eq. (8) reconstruct $\hat Z_{t+1:t+P}$ during inference without already having access to the future region tokens? A precise tensor-level explanation would improve both soundness and presentation.
3. **Can the authors provide actual efficiency measurements, not only asymptotic complexity?**
   I would like to see the preprocessing time for affinity construction and spectral clustering, as well as training and inference runtime, especially for multi-step autoregressive rollout, compared against PatchSTG or another strong baseline. This would help improve the paper's significance.
4. **What is the main reason NeST outperforms PatchSTG?**
   The paper should more clearly isolate whether the gain primarily comes from semantic clustering, future guidance, cross-scale attention, or the combined training objective. This would help clarify the paper’s core contribution and originality.

**Limitations:**

The paper includes an impact statement, but the discussion of limitations is not yet adequate. It should more explicitly discuss dependence on clustering quality, the cost of region construction, the consequences of using one global clustering assignment for all time steps, the training-inference gap introduced by future guidance reconstruction, and the practical overhead of autoregressive rollout.

**Strengths And Weaknesses:**

**Strengths**

- The paper constructs the node affinity matrix directly from data, rather than relying on a predefined graph, and then uses spectral clustering to derive region-level auxiliary series. This is a interesting and reasonable way to discover similarity structure from the data itself and to build a coarse-grained representation for forecasting.
- A notable idea in the paper is to use future coarse-grained regional representations as guidance for fine-grained node forecasting. From a prediction perspective, this allows the model to leverage region-level future trends as a relatively stable structural signal, and then interact with node-level sequences through cross-scale attention with near-linear complexity. This is a meaningful design choice for extracting spatio-temporal features and correcting node-level forecasting bias.
- The empirical study is fairly extensive, and the reported results on multiple large-scale datasets are strong. The paper includes comparisons with a substantial set of baselines, as well as ablation and long-horizon analyses, which together suggest that the proposed framework is effective in practice.

However, the paper also has several notable weaknesses that need to be addressed before the contribution can be fully assessed, especially in terms of methodological clarity, consistency of the technical description, efficiency analysis, reproducibility, and discussion of limitations.


**Weaknesses**

- **The presentation of the core pipeline is not sufficiently clear.**

  - In Section 4.1, the clustering pipeline, especially the “spectral representation” part, is still not described concretely enough in the main text. Although Appendix Algorithm 1 appears to clarify the process, the main paper should itself explain more explicitly how the affinity matrix is turned into spectral embeddings, how clustering is performed, and how the final region-level series are obtained.
  - Section 4.2 is also difficult to follow, with several under-specified definitions and notation issues.

    - First, $TE$ is introduced only as a learnable embedding, but its role and dimensionality are not clearly specified. Since $X$ and $Z$ are already flattened and projected into $N \times d$ and $M \times d$ latent tokens, the manuscript should explicitly state the dimensions and implementation of $TE_x$ and $TE_z$.
    - Second, $H_z^{zeros}$ in Eq. (8) is not properly defined. More importantly, the exposition creates a potential circular dependency: $\tilde H_x$ in Eq.(8) (for reconstructing $\hat Z_{t+1:t+P}$ ) is computed via attention with future region guidance $H_z^{fut}$, yet $H_z^{fut}$ depends on $Z_{t+1:t+P}$, which is unavailable at inference time. Although the paper later says that inference initializes guidance with zero masks and reconstructs $\hat Z_{t+1:t+P}$, the exact sequence of operations is still not made sufficiently explicit, which makes the inference pipeline confusing.
    - Third, Algorithm 2 is helpful as a summary, but the same process is not expressed clearly in the main text, and even Algorithm 2 itself uses notation that is insufficiently explained. For example, it is unclear whether $u$ and $z$ denote sequences or latent representations, whether $u_{t+1}$ and $z_{t+1}$ are the same object, and how this index system maps back to the paper’s main notation such as $Z_{t+1:t+P}$.
- **The claimed efficiency story is incomplete.**  The paper emphasizes near-linear cross-attention complexity, $O(lNMd)$, which is a meaningful point, but this analysis excludes the cost of building the temporal affinity matrix, computing the spectral representation, performing eigen-decomposition, and running K-means clustering, all of which are central to the method. Since the paper explicitly targets large-scale settings, these preprocessing costs should be analyzed more explicitly.
- **The paper also does not compare the runtime cost of multi-step autoregressive rollout against relevant baselines.**  This is important because NeST performs future-region reconstruction, then recurrent regional guidance generation, and repeated rollout over patches. The practical inference overhead should therefore be reported. More broadly, given that the forecasting setting effectively involves predicting two future segments and then rolling forward autoregressively, the paper should describe much more clearly: (1) how training samples are constructed, and where exactly autoregression enters during training, and (2) at test time, when future values are unavailable, what the full input-output pipeline is step by step. As currently written, it is not yet sufficiently clear whether there is any risk of information leakage in how future regional guidance is used during training versus inference.
- **There are also several presentation issues in figures and descriptions.**  In Figure 1, the right-side label "$\hat{Z}_{t+P+1:t+2}$" appears to contain an indexing mistake. In Section 5.4, the case-study description refers to “Region 547 (Orange),” but this is not visually consistent with the actual figure.
- **Reproducibility is weaker than it should be.**  Although some hyperparameters are provided in Appendix A.5, several important quantities are still not clearly instantiated in the main paper, including the kernel bandwidth $\sigma$, the scheduled-sampling decay settings $(\gamma, r)$, and the exact specification of $TE$. I also did not find a code release statement in the submission.
- The limitations discussion is still too light relative to the method. The submission should more directly discuss dependence on clustering quality, sensitivity to preprocessing and affinity construction choices, preprocessing overhead, and the limitations of using a single clustering assignment shared across time.

---

> ### Author Rebuttal · Authors · 2026-03-31
>
> # Response to Reviewer fSep
> We thank the reviewer for the feedback.
>
> ## [W2, Q1, Q2] Training & Inference Pipeline and Symbols
> We apologize for the notation ambiguities and confirm no circular dependency or future leakage.
>
> To clarify, let $\mathcal{M}$ denote our core network (Encoding + Cross-Attention):
> $(\tilde{\mathbf{H}}\_x, \tilde{\mathbf{H}}\_z) = \mathcal{M}(\mathbf{X}\_{t}, \mathbf{Z}\_{t+1})$.
> Our causal pipeline uses three decoupled heads: $\text{Proj}\_x$ (predicts nodes), $\text{Proj}\_z$ (predicts next macro-step), and $\text{Proj}\_{\rm bd}$ (reconstructs guidance purely from history). For $L=1, P=1$, the data flow is:
> 1. **Training Phase (Scheduled Sampling)**:
> - **Teacher forcing** (prob $p$): Uses future macro-trend $\mathbf{Z}\_{t+1}$.
> $(\tilde{\mathbf{H}}\_x, \tilde{\mathbf{H}}\_z) = \mathcal{M}(\mathbf{X}\_t, \mathbf{Z}\_{t+1}) \implies \hat{\mathbf{X}}\_{t+1} = \text{Proj}\_x(\tilde{\mathbf{H}}\_x), \hat{\mathbf{Z}}\_{t+2} = \text{Proj}\_z(\tilde{\mathbf{H}}\_z)$
> - **Autoregressive simulation** (prob $1-p$): Simulates inference by zero-masking.
> $(\\_ , \tilde{\mathbf{H}}'\_z) = \mathcal{M}(\mathbf{X}\_t, \mathbf{0}) \implies \hat{\mathbf{Z}}\_{t+1} = \text{Proj}\_{\rm bd}(\tilde{\mathbf{H}}'\_z)$
> - - Joint Optimization: Optimizes reconstruction ($\hat{\mathbf{Z}}\_{t+1}$), node prediction ($\hat{\mathbf{X}}\_{t+1}$), and rollout ($\hat{\mathbf{Z}}\_{t+2}$)
>
> 2. **Inference phase**:
> - Step 1 (**Bootstrap**): Reconstruct guidance via masking.
> $(\\_ , \tilde{\mathbf{H}}'\_z) = \mathcal{M}(\mathbf{X}\_t, \mathbf{0}) \implies \hat{\mathbf{Z}}\_{t+1} = \text{Proj}\_{\rm bd}(\tilde{\mathbf{H}}'\_z)$
> - Step 2 (**Predict**): Generate nodes and the next macro-step from $\hat{\mathbf{Z}}\_{t+1}$.
> $(\tilde{\mathbf{H}}\_x, \tilde{\mathbf{H}}\_z) = \mathcal{M}(\mathbf{X}\_t, \hat{\mathbf{Z}}\_{t+1}) \implies \hat{\mathbf{X}}\_{t+1} = \text{Proj}\_x(\tilde{\mathbf{H}}\_x), \hat{\mathbf{Z}}\_{t+2} = \text{Proj}\_z(\tilde{\mathbf{H}}\_z)$
> - Step 3 (**Rollout**): Feed $\hat{\mathbf{Z}}\_{t+2}$ into the next time step and repeat Step 2.
>
> Symbol definitions (Q2):
> - $\mathbf{TE}\_x, $\mathbf{TE}\_z$: Learnable Time-of-Day and Day-of-Week embeddings, averaged across the horizon.
> - $\mathbf{H}\_{z}^{\rm zeros}$: Hidden state encoded from an all-zero mask, capturing the prior state before cross-scale interaction.
>
> ## [W3, Q3] Efficiency Measurements
> We evaluated the empirical runtime on the GBA dataset using a single RTX 4090 GPU (batch size 64).
>
> |Phase | Metric | PatchSTG | NeST (ours) |
> |--- | --- | --- | --- |
> | Preprocessing| Affinity Construction| N/A | 22 s|
> | | Spectral Clustering| N/A| 69s |
> | Training| Time/Epoch| 185s | 75s|
> | Inference | Time | 32s | 20s|
>
> Despite the one-time offline preprocessing overhead (91s), our **linear cross-attention** and **patch-wise prediction** designs bound computational complexity.
>
> ## [Q4] Source of Gain vs. PatchSTG
> The advantage of NeST over PatchSTG is a paradigm shift: PatchSTG predicts long-term futures directly, while NeST leverages macro future guidance as anchors. Ablations confirm this synergy drives our gains:
>
> - **Semantic Clustering**: Random/predefined partitioning misses latent correlations, causing a more than 2\% MAE drop (see **Table 4**).
> - **Future Guidance**: Replacing future guidance $\mathbf{Z}\_{t+1}$ with historical $\mathbf{Z}\_t$ causes a 1.71\% MAE degradation (18.73 $\to$ 19.05 on GBA), highlighting its role in stabilizing node-level trajectories.
> - **Cross-Scale Attention**: Complete removal causes severe degradation (see **Table 3**); even replacing it with naive concatenation+MLP drops MAE by 1.71\%.
>
> ## [W1] Clarity of the Clustering Pipeline
> We will clarify this in the revised main text with the following concrete steps:
> - **Daily Chunking**: Partition training $\mathbf{X}$ into non-overlapping daily chunks to capture periodicity.
> - **Affinity Matrix** ($\mathbf{A} \in \mathbb{R}^{N \times N}$): Apply a Gaussian kernel on Euclidean distances between chunks.
> - **Spectral Embedding** ($\mathbf{V} \in \mathbb{R}^{N \times k}$): Extract the $k$ smallest eigenvectors of the normalized Laplacian $\mathbf{L}\_{\rm sym} = \mathbf{I} - \mathbf{D}^{-1/2} \mathbf{A} \mathbf{D}^{-1/2}$.
> - **Aggregation** ($\mathbf{Z}$): K-Means on rows of $\mathbf{V}$ yields assignment matrix $\mathbf{S}\in \{0,1\}^{N\times M}$, grouping nodes to aggregate signals into macro sequences $\mathbf{Z}$.
>
> ## [W4-W6] Presentation, Reproducibility, and Limitations
> We appreciate your meticulous review and will update the manuscript accordingly:
> - **Presentation (W4)**: Thank you! We will fix the typos.
> - **Reproducibility (W5)**: We will detail all hyperparameters (e.g., kernel bandwidth, decay settings) and open-source our code, pre-trained weights, and scripts upon acceptance.
> - **Limitations (W6)**: We will expand discussions on clustering quality, affinity sensitivity, preprocessing overhead, and the limits of a single static clustering assignment over time.

---

> > ### Author Rebuttal · Reviewer_fSep · 2026-04-01
> >
> > Thank you for the response. I recognize the paper’s novelty and experimental strength, and I believe the planned improvements in presentation will make it a good paper. I have increased my score.

---

> > > ### Author Response · Authors · 2026-04-02
> > >
> > > Dear Reviewer fSep,
> > >
> > > We sincerely thank you for your encouraging response and the positive score adjustment! Your constructive suggestions have been very helpful, and we are fully committed to incorporating them into the final revision. Thank you!
> > >
> > > Authors

---

### Official Review · Reviewer_HkQv · 2026-03-11

**Soundness:** 2
**Presentation:** 2
**Significance:** 3
**Originality:** 3
**Overall Recommendation:** 4
**Confidence:** 4

**Summary:**

This paper proposes NEST, a novel nested spatio-temporal forecasting framework that addresses the challenge of noise sensitivity in fine-grained predictions. Its core contribution is a future-aware, macro-guided paradigm consisting of two key innovations: 1) employing semantic spectral clustering on raw temporal data to dynamically group nodes into coherent regions, extracting de-noised, representative centroid features that preserve systematic trends; and 2) introducing a nested architecture where predicted future macro-level regional states serve as explicit top-down guidance to regularize and enhance micro-level node forecasting through a bidirectional cross-scale attention mechanism. Extensive experiments on large-scale traffic datasets demonstrate that NEST consistently outperforms state-of-the-art baselines, achieving more accurate and robust long-horizon predictions.

**Compliance With Llm Reviewing Policy:**

Affirmed.

**Final Justification:**

I think the author's rebuttal has solve the weaknesses well. Although the innovation of Nest is not so outstanding, but the experiments are very detailed. So I think it's worthy to give a weak accept to this paper.

**Key Questions For Authors:**

1. While the paper clearly presents its motivation and methodology, its experimental comparison is criticized as unfair for omitting benchmarks against specialized models robust to noise and non-stationarity.
2. Although it demonstrates advantages on large-scale traffic datasets, the claims of robustness and generalization are limited due to evaluations on homogeneous traffic data alone, see in W2.
3. The core "future macro-guidance" paradigm is not entirely novel, as similar ideas exist in related fields; its primary innovation lies in the specific adaptation to spatio-temporal graph data.
4. The work's soundness is further weakened by the lack of in-depth visualization for the key "centroid features," which could have intuitively demonstrated the noise-filtering process.

**Strengths And Weaknesses:**

1. The paper's motivation and methodology are clearly presented. However, it is quite appalling that the experimental section does not include comparisons with existing spatio-temporal forecasting methods specifically designed to be robust against noisy and non-stationary conditions. Instead, it only conducts direct comparisons with standard forecasting models, which is evidently unfair.
2. Experiments are conducted on three large-scale traffic datasets (GBA, GLA, CA), demonstrating performance advantages. However, the evaluation scenarios are relatively homogeneous (all are PeMS traffic flow data). The paper does not test on more complex, differently noised, or more dynamic scenarios, so its claims of "robustness" and "generalization ability" are primarily confined to the traffic domain.
3. The core idea is "using predicted macro-level future states to guide micro-level forecasting". However, this paradigm of "future macro-guidance" has been explored in related fields (e.g., video prediction, some hierarchical modeling works). The innovation seems lie more in the specific implementation tailored for spatio-temporal graph data, combining spectral clustering with a dedicated interaction mechanism, rather than in the absolute originality of the paradigm itself.
4. The paper lacks an in-depth visual analysis of the "centroid features," a core intermediate variable. Such an analysis could have more intuitively revealed how the method filters noise and preserves trends in the temporal dimension, thereby strengthening the argument's persuasiveness.

---

> ### Author Rebuttal · Authors · 2026-03-31
>
> # Response to Reviewer HkQv
>
> We thank the reviewer for your constructive feedback.
>
> ## [Q1 \& W1] Comparison with Specialized Robust/Non-stationary Models
> We respectfully clarify that NeST targets **general spatio-temporal forecasting** rather than exclusively addressing noise or non-stationarity, which is why we primarily evaluated against general-purpose models (e.g., PatchSTG). However, to empirically demonstrate inherent robustness, we additionally evaluated NeST on the GBA dataset against **STMPNet** [1], a dedicated probabilistic model explicitly designed to mitigate local noise and capture uncertain distributions:
>
> |Model|MAE|RMSE|MAPE|
> |---|---|---|---|
> |STMPNet| 23.75|36.96 | 19.31 |
> |**NeST(ours)**|18.73|31.85|12.90|
>
> NeST substantially outperforms this specialized baseline, confirming that stable regional soft guidance naturally mitigates noise without complex probabilistic modules.
>
> [1] Spatio-Temporal Multivariate Probabilistic Modeling for Traffic Prediction, 2025, TKDE.
>
> ## [Q2 & W2] Generalization to Non-traffic Domains
> To verify NeST's applicability beyond traffic data, we evaluated our framework on two distinct non-traffic spatio-temporal domains (**KnowAir** for Meteorology, **UrbanEV** for Energy) and two classical time series benchmarks (**Electricity, Solar-Energy**).
>
> **1. Non-Traffic Spatio-Temporal Forecasting**
>
> | Model | KnowAir (MAE/RMSE) | UrbanEV (MAE/RMSE) |
> | :--- | :--- | :--- |
> | Air-DualODE [1] | 18.64 / 29.37 | - |
> | PatchSTG | 16.08 / 24.70  | 5.16 / 11.53  |
> | MAGE [2] | 15.36 / 23.42  | 4.95 / 11.00 |
> | **NeST (Ours)** | **14.87** / **22.89**| **4.37** / **10.26** |
>
> **2. Classical Long-Term Time Series Forecasting (96$\rightarrow$96)**
>
> | Model | Electricity (MSE/MAE) | Solar-Energy (MSE/MAE) |
> | :--- | :--- | :--- |
> | PatchTST [3]| 0.190 / 0.296 | 0.265 / 0.323 |
> | iTransformer [4] | 0.148 / 0.240 | 0.203 / 0.237 |
> | TimeMixer [5] | 0.153 / 0.247 | **0.189** / 0.259 |
> | **NeST (Ours)** | **0.141** / **0.235** | 0.201 / **0.211** |
>
> These results demonstrate NeST's robust generalization capabilities across diverse domains, consistently outperforming both recent SOTA spatio-temporal models and leading temporal transformers. \\\
>
> References:
> [1] MAGE: Less but More: Linear Adaptive Graph Learning
> Empowering Spatiotemporal Forecasting, NeurIPS, 2025.
>
> [2] Air-DualODE: Air Quality Prediction with Physics-Guided Dual Neural ODEs in Open Systems, ICLR, 2025.
>
> [3] iTransformer: Inverted Transformers Are Effective for Time Series Forecasting, 2024, ICLR
>
> [4] PatchTST: A Time Series is Worth 64 Words: Long-term Forecasting with Transformers, 2023 ICLR.
>
> [5] TimeMixer: Decomposable Multiscale Mixing for Time Series Forecasting, ICLR 2024.
>
> ## [Q3 & W3] Novelty
> While a high-level concept of macro-guidance exists in domains such as video generation, we emphasize that the non-Euclidean topology and evolving dynamics introduces fundamental challenge, which requires both theoretical motivation and algorithmic redesign.
>
> First, instead of relying a a static hierarchy, we extract dynamics-aware latent feature with spectral clustering, operating on affinity matrix that is defined using temporal differences. This directly contrasts to existing approaches based on a simple downsampling strategy or fixed prior knowledge.
> Second, as far as we are concerned, we are the first to propose the future-aware guidance in irregular spatiotemporal domains. Prior works, even considering the methods for video generation, are limited to regular grids with standard backbones (ViT/UiT-based architectures). Our design of the the bi-level attention allows us to formulate the regional trends with affordable computational cost.
>
> Empirically, our proposed architecture yields significant improvements across multiple datasets compared to recent state-of-the-art baselines, firmly validating the practical value of our paradigm.
>
> ## [Q4 & W4] Visualization of Centroid Features and Noise-filtering
> We provide a visual analysis (see [Anonymous](https://anonymous.4open.science/r/ICML_id15811/README.md)) comparing raw node sequences with their corresponding macro centroid features. This visualization explicitly demonstrates how our clustering acts as a **natural low-pass filter**: while individual nodes exhibit **high-frequency noise** and sharp fluctuations, macro centroids extract and preserve **smooth trends**. These **stable regional anchors** allow the cross-scale attention to robustly correct noisy micro-predictions. This intuitive evidence confirms our architectural intuition: leveraging macro-level consistency to shield node-level forecasting from **localized anomalies**.

---

> > ### Author Rebuttal · Reviewer_HkQv · 2026-04-02
> >
> > I think the author's rebuttal has solve the weaknesses well. Although the innovation of Nest is not so outstanding, but the experiments are very detailed. So I think it's worthy to give a weak accept to this paper. Good luck to the authors.

---

> > > ### Author Response · Authors · 2026-04-03
> > >
> > > Dear reviewer HkQv,
> > >
> > > Thank you very much for your positive confirmation and recommendation! We truly appreciate your recognition and are happy to know that our rebuttal resolved your core concerns. We will thoroughly polish the final revision based on your feedback. Thank you!
> > >
> > > Authors

---

### Official Review · Reviewer_opEH · 2026-03-12

**Soundness:** 3
**Presentation:** 2
**Significance:** 2
**Originality:** 2
**Overall Recommendation:** 3
**Confidence:** 5

**Summary:**

NeST is a two-level forecasting framework for graph-structured spatio-temporal data The macro level uses spectral clustering to discover latent node regions with shared dynamics, produces coarse region-level forecasts, and feeds them as "future guidance" to the micro level, which makes fine-grained per-node predictions. The design includes cross-scale attention, teacher forcing with scheduled sampling, and quantile regression for uncertainty-aware guidance. Experiments on three large-scale traffic datasets (GBA, GLA, CA from LargeST) show improvements over baselines.

**Compliance With Llm Reviewing Policy:**

Affirmed.

**Final Justification:**

I do not recommend acceptance at this time. In my view, the work does not demonstrate sufficiently broad impact for the machine learning community.

**Key Questions For Authors:**

1. Is the spectral clustering done once at the start or periodically updated? If the underlying traffic patterns shift (e.g., new road construction), the clusters could become stale.

2. What do the discovered regions look like on a map? Are they geographically contiguous, or do they group distant sensors with similar patterns?

3. How does the model handle the case where macro guidance is actually misleading, e.g., a regional trend suggests increasing traffic but a specific node has a local event causing a decrease?

**Limitations:**

This is a traffic-specific paper dressed up as a general spatio-temporal method.  I don't think it can serve the ML/AI community.

**Strengths And Weaknesses:**

## Strengths

- The nested coarse-to-fine idea is sound. It makes sense to first capture these aggregate trends before trying to predict individual sensors.

- The uncertainty-aware guidance via quantile regression is a nice touch. Point estimates for future macro trends would be fragile; passing distributional information to the micro level adds robustness.

- The long-horizon stability analysis is appreciated. Many traffic models look good at short horizons but fall apart further out. NeST maintains reasonable performance at 48 steps.

## Weaknesses

- The evaluation is exclusively on traffic data, and all three datasets come from the same source (LargeST, California road sensors). This is a major weakness. Traffic has very specific characteristics, such as strong periodicity, spatial locality along road networks, similar sensor types. I have no idea if NeST works for other spatiotemporal fields.

- The number of regions M is set to 0.2N based on what looks like manual tuning. The sensitivity analysis shows a U-shaped curve with significant degradation at 0.1N and 0.3N. This is concerning, as a 2x change in M causes noticeable performance drops.

- Teacher forcing with scheduled sampling is a known recipe for exposure bias. The paper doesn't measure how much the inference behavior degrades due to the train-test discrepancy. At longer horizons (12+ steps), the autoregressive rollout could compound errors from the macro level that weren't seen during training.

- Baselines are too outdated. In my view, there have been many more advanced deep learning methods for traffic forecasting proposed in 2024–2025. There are also traffic foundation models from scratch and LLM-based approaches.

---

> ### Author Rebuttal · Authors · 2026-03-31
>
> We thank the reviewer for your feedback.
>
> ## [Q1] Evolving Graph Topologies
>
> While handling evolving topologies is an important problem in practice, all reported baselines are evaluated with static spatial structure, and we choose this setup for fair comparison.
> However, NeST can adapt to real-world shifts.
> Specifically, the affinity matrix that the spectral clustering operates on is defined on patch-level temporal differences, allowing it to adapt to spatio-temporal structure in the evloving dynamics.
> Furthermore, since our preprocessing introduces minimal overhead (91s), periodically re-executing the offline clustering can be realized to capture shifted patterns, where the updated group assignments can then be encoded via positional encoding, making the trained model adjust to the evloving patterns during the inference.
>
> ## [Q2] Geographic vs. Semantic Region Discovery
> NeST uses data-driven semantic clustering based on historical pattern consistency, not strict geography. While adjacent sensors naturally group together, NeST transcends spatial proximity to capture long-range semantic correlations, integrating both local contiguity and distant functional semantics for comprehensive regional trends.
>
> ## [Q3] Robustness to Misleading Macro Guidance
> NeST addresses misleading macro guidance by treating regional trends as soft conditions. When a node experiences a local anomaly contradicting the macro trend, cross-scale attention computes a low similarity, dynamically downweighting the conflicting macro signal. Besides, residual connections preserve distinct local information, maintaining stability during localized anomalies.
>
> ## [W1] Generalization to Non-traffic Domains
> We evaluated NeST on non-traffic spatio-temporal domains (**KnowAir** for Meteorology, **UrbanEV** for Energy) and classical time series benchmarks (**Electricity, Solar-Energy**).
>
> **1. Non-Traffic Spatio-Temporal Forecasting**
>
> | Model | KnowAir (MAE/RMSE) | UrbanEV (MAE/RMSE) |
> | :--- | :--- | :--- |
> | Air-DualODE [1] | 18.64 / 29.37 | - |
> | PatchSTG | 16.08 / 24.70  | 5.16 / 11.53  |
> | MAGE [2] | 15.36 / 23.42  | 4.95 / 11.00 |
> | **NeST (Ours)** | **14.87** / **22.89**| **4.37** / **10.26** |
>
> **2. Classical Long-Term Time Series Forecasting (96$\rightarrow$96)**
>
> | Model | Electricity (MSE/MAE) | Solar-Energy (MSE/MAE) |
> | :--- | :--- | :--- |
> | PatchTST [3]| 0.190 / 0.296 | 0.265 / 0.323 |
> | iTransformer [4] | 0.148 / 0.240 | 0.203 / 0.237 |
> | TimeMixer [5] | 0.153 / 0.247 | **0.189** / 0.259 |
> | **NeST (Ours)** | **0.141** / **0.235** | 0.201 / **0.211** |
>
> These results demonstrate NeST's robust generalization capabilities across diverse domains, consistently outperforming both recent SOTA spatio-temporal models and leading temporal transformers.
>
> References:
> [1] MAGE: Less but More: Linear Adaptive Graph Learning Empowering Spatiotemporal Forecasting, NeurIPS, 2025.
>
> [2] Air-DualODE: Air Quality Prediction with Physics-Guided Dual Neural ODEs in Open Systems, ICLR, 2025.
>
> [3] iTransformer: Inverted Transformers Are Effective for Time Series Forecasting, 2024, ICLR
>
> [4] PatchTST: A Time Series is Worth 64 Words: Long-term Forecasting with Transformers, 2023 ICLR.
>
> [5] TimeMixer: Decomposable Multiscale Mixing for Time Series Forecasting, ICLR 2024.
>
> ## [W2] Sensitivity Analysis of Region Number ($M$)
> Detailed analysis of $M \in [0.1N, 0.3N]$ on the GBA dataset:
>
> |M|0.1N|0.15N|0.18N|0.20N|0.22N|0.25N|0.30N|
> |---|---|---|---|---|---|---|---|
> |MAE|19.06|19.03|18.89|18.73|18.96|19.14|19.29|
>
> While altering spatial granularity creates a tradeoff between oversmoothing and overfragmenting, results reveal a **stable performance plateau** around $0.2N$. Crucially, even degraded performance at suboptimal boundaries ($0.1N$, $0.3N$) still outperforms PatchSTG (19.50), confirming inherent robustness without strict manual tuning.
>
> ## [W3] Train-Test Discrepancy
> To prevent cascading macro-level errors, NeST employs two defenses:
>
> - **Autoregressive Simulation**: Scheduled sampling exposes the model to prediction errors during training.
> - **Soft Conditioning**: If macro trends accumulate errors, cross-scale attention dynamically downweights them, safely falling back on local features via residuals.
>
> Empirically, our 12-to-48 step results (Table 2) show consistent performance, confirming NeST effectively mitigates exponential error compounding.
>
> ## [W4] Timeline of baselines and LLM-based approaches
> We compared NeST against contemporary baselines **PRMixer (2024) and PatchSTG (2025)**. To further address your concern, we added comprehensive evaluations **MAGE (NeurIPS 2025), Air-DualODE (ICLR 2025), TimeMixer (ICLR 2024), iTransformer (ICLR 2024) and PatchTST (ICLR 2023)** in **[W1]**, where NeST maintains a clear advantage.
>
> Regarding LLMs, they represent an orthogonal paradigm dependent on massive external data and compute. In contrast, NeST is a **lightweight, highly efficient** architecture trained from scratch.

---

> > ### Author Rebuttal · Reviewer_opEH · 2026-04-03
> >
> > I have updated my score to weak reject, but I still do not recommend acceptance at this time. In my view, the work does not demonstrate sufficiently broad impact for the ML community.

---

> > > ### Author Response · Authors · 2026-04-07
> > >
> > > Dear Reviewer opEH,
> > >
> > >
> > > We sincerely thank you for your continued engagement and thoughtful feedback. While we appreciate your perspective, we respectfully disagree with your assessment regarding the broader impact of our work. Our work addresses a fundamental challenge in spatio-temporal forecasting: how to accurately model high-dimensional, long-horizon dynamics without relying on predefined physical priors. We believe this question is of broad relevance across multiple domains, and our contributions advance it along three key dimensions:
> > >
> > >
> > > **(1) Methodological contribution:** We proposed a nested forecasting framework with future-aware guidance, specifically designed for irregular and high-dimensional spatio-temporal data. Unlike existing approaches that depend on predefined graphs or static spatial priors, our method constructs data-driven semantic regions, enabling adaptive alignment with evolving macro dynamics while maintaining  computational efficiency.
> > >
> > >
> > > **(2) Theoretical insight:** We showed that our formulation of affinity matrix, defined through patch-wise temporal differences, induces a representation that is inherently robust to noise and can adaptively capture the dynamic structures, leading to stability and better perservation of region-level trends over long-horizons.
> > >
> > >
> > > **(3) Empirical generalization:** We evaluated the proposed framework across diverse domains, including traffic (e.g., GBA, GLA from the LargeST benchmark [1]), meteorology (KnowAir [2]), energy (UrbanEV [3]), and classical time series benchmarks (Electricity, Solar-Energy [4]). The results consistently show improvements over an extensive set of baselines, ranging from established top-tier methods (e.g., DSTAGNN, ICML 2022 [5]; PatchTST, ICLR 2023 [6]; TimeMixer, ICLR, 2024) to the latest recent spatio-temporal models (e.g., PatchSTG, KDD 2025 [7]; MAGE, NeurIPS 2025 [8]; Air-DualODE, ICLR 2025 [9]) and strong temporal baselines (e.g., iTransformer, ICLR 2024 [10]). These experiments aim to demonstrate that the proposed paradigm is not tied to a specific application domain, but generalizes across different types of spatio-temporal data.
> > >
> > >
> > > We appreciate if you could continue your engagement with us. We thank you again for your time and consideration.
> > >
> > >
> > > Authors
> > >
> > >
> > > References:
> > >
> > >
> > > [1] LargeST: A Benchmark Dataset for Large-Scale Traffic Forecasting, **NeurIPS, 2023.**
> > >
> > > [2] PM2.5-GNN: A Domain Knowledge Enhanced Graph Neural Network For PM2.5 Forecasting, IGSPATIAL, 2020.
> > >
> > > [3] UrbanEV: An Open Benchmark Dataset for Urban Electric Vehicle Charging Demand Prediction, **Scientific Data, 2025.**
> > >
> > > [4] Modeling Long- and Short-Term Temporal Patterns with Deep Neural Networks, **SIGIR, 2018.**
> > >
> > > [5] DSTAGNN: Dynamic Spatial-Temporal Aware Graph Neural Network for Traffic Flow Forecasting, **ICML, 2022.**
> > >
> > > [6] A Time Series is Worth 64 Words: Long-term Forecasting with Transformers, **ICLR, 2023.**
> > >
> > > [7] Efficient Large-Scale Traffic Forecasting with Transformers: A Spatial Data Management Perspective, **KDD, 2025.**
> > >
> > > [8] MAGE: Less but More: Linear Adaptive Graph Learning Empowering Spatiotemporal Forecasting, **NeurIPS, 2025.**
> > >
> > > [9] Air-DualODE: Air Quality Prediction with Physics-Guided Dual Neural ODEs in Open Systems, **ICLR, 2025.**
> > >
> > > [10] iTransformer: Inverted Transformers Are Effective for Time Series Forecasting, **ICLR, 2024**.
> > >
> > > [11] TimeMixer: Decomposable Multiscale Mixing for Time Series Forecasting, **ICLR, 2024.**

---

### Official Review · Reviewer_dTqY · 2026-03-13

**Soundness:** 4
**Presentation:** 3
**Significance:** 3
**Originality:** 3
**Overall Recommendation:** 4
**Confidence:** 3

**Summary:**

This paper proposes NEST, a framework that leverages macro-level trends to guide fine-grained node-level predictions. The framework first groups nodes into semantic regions using spectral clustering, models regional temporal dynamics, and then integrates region-level trends with node representations through cross-scale attention. By incorporating macro-level guidance into node-level forecasting, the approach aims to improve prediction accuracy and stability. Overall, the proposed solution is sound and reasonable.

**Compliance With Llm Reviewing Policy:**

Affirmed.

**Final Justification:**

This paper develops  a new framework that leverages macro-level trends to guide fine-grained node-level predictions. By incorporating macro-level guidance into node-level forecasting, the approach aims to improve prediction accuracy and stability. Overall, the proposed solution is sound and well evaluated. The rebuttal has addressed my major concerns.

**Key Questions For Authors:**

(1) While the paper provides theoretical complexity analysis, could the authors further discuss the running time of NEST compared with baselines?
(2) The paper studies the number of regions M using three settings (0.1N, 0.2N, and 0.3N). Could the authors provide a more fine-grained analysis of this hyperparameter to better understand its impact on performance?
(3) How does the model compare with recent transformer-based forecasting architectures that do not rely on graph structures?
(4) Can the proposed framework generalize to other spatio-temporal forecasting tasks (e.g., weather or energy forecasting)? Have the authors tested the model outside traffic datasets?

**Limitations:**

The paper does not provide sufficient discussion of the limitations of the proposed framework. In particular, the method is evaluated mainly on traffic forecasting datasets, and its applicability to other spatio-temporal forecasting tasks remains to be further investigated.

**Strengths And Weaknesses:**

Strengths
1. The idea of using macro-level regional dynamics to guide node-level forecasting is interesting.
2. Experiments on large-scale datasets demonstrate the effectiveness of the proposed framework.
3. Overall, this paper is well written and easy to follow.

Weaknesses
1. The technical contribution is not very significant. The novelty is somewhat limited as the framework mainly integrates existing approaches.
2. The region construction relies on static clustering and is not jointly optimized with the model.
3. The paper provides a limited analysis of how regional dynamics influence node-level predictions.

---

> ### Author Rebuttal · Authors · 2026-03-31
>
> # Response to Reviewer dTqY
> We thank the reviewer for constructive feedback!
>
> ## [Q1] Running Time and Efficiency
> Empirical runtime vs. PatchSTG on an RTX 4090 (batch 64):
>
> |Dataset|Model| Training Time (s/epoch) | Inference Time (s)|
> |---|---|---|---|
> | GBA | PatchSTG | 185 | 32 |
> | |NeST(ours) | 75 | 20 |
> | GLA| PatchSTG| 235 | 42 |
> | | NeST (ours) | 137 | 37 |
>
> As theorized, NeST markedly speeds up both training and inference stages.
> Empirical results confirm our linear cross-attention and patch-wise prediction approaches effectively constrain computational costs.
>
> ## [Q2] Fine-grained Analysis of Region Number ($M$)
> Detailed analysis of region count $M \in [0.1N, 0.3N]$ on GBA dataset is provided:
>
> |M|0.1N|0.15N|0.18N|0.20N|0.22N|0.25N|0.30N|
> |---|---|---|---|---|---|---|---|
> |MAE|19.06|19.03|18.89|18.73|18.96|19.14|19.29|
>
> The U-shaped curve reflects a spatial trade-off between over-smoothing (small $M$) and over-fragmenting (large $M$).
> Crucially, NeST consistently **surpasses PatchSTG** (19.50), even when using sub-optimal choices of $M$ (0.1N, 0.3N), implying its robustness over a broad range.
>
> ## [Q3] Comparison with recent transformer-based forecasting architectures
> Additional experiments on GBA using competitive transformer-based time-series forecasting baselines.
>
> |Model|MAE|RMSE|
> |---|---|---|
> |iTransformer [1]| 46.56| 74.55|
> |PatchTST [2]| 36.24| 60.46 |
> |NeST |18.73 | 31.85 |
>
> While iTransformer and PatchTST are highly competitive in modeling temporal dependencies, these results suggest temporal modeling alone is insufficient for spatio-temporal data. Notably, although NeST does not explicitly rely on predefined graph structures, it still significantly outperforms these baselines by capturing spatial dependencies through its macro-to-micro design.
>
> [1] iTransformer: Inverted Transformers Are Effective for Time Series Forecasting, 2024, ICLR
>
> [2] PatchTST: A Time Series is Worth 64 Words: Long-term Forecasting with Transformers, 2023 ICLR.
>
> ## [Q4] Experiments on Non-traffic data
>
> To verify applicability beyond traffic, we evaluated NeST on two non-traffic spatio-temporal domains (**KnowAir** for Meteorology, **UrbanEV** for Energy) and two classical time series benchmarks (**Electricity, Solar-Energy**).
>
> **1. Non-Traffic Spatio-Temporal Forecasting**
>
> | Model | KnowAir (MAE/RMSE) | UrbanEV (MAE/RMSE) |
> | :--- | :--- | :--- |
> | Air-DualODE [1] | 18.64 / 29.37 | - |
> | PatchSTG | 16.08 / 24.70  | 5.16 / 11.53  |
> | MAGE [2] | 15.36 / 23.42  | 4.95 / 11.00 |
> | **NeST (Ours)** | **14.87** / **22.89**| **4.37** / **10.26** |
>
> **2. Classical Time Series Forecasting (96$\rightarrow$96)**
>
> | Model | Electricity (MSE/MAE) | Solar-Energy (MSE/MAE) |
> | :--- | :--- | :--- |
> | PatchTST | 0.190 / 0.296 | 0.265 / 0.323 |
> | iTransformer | 0.148 / 0.240 | 0.203 / 0.237 |
> | TimeMixer [3] | 0.153 / 0.247 | **0.189** / 0.259 |
> | **NeST (Ours)** | **0.141** / **0.235** | 0.201 / **0.211** |
>
> These results demonstrate NeST's robust cross-domain generalization, consistently outperforming both recent SOTA spatio-temporal models and leading temporal transformers.
>
>
> [1] MAGE: Less but More: Linear Adaptive Graph Learning
> Empowering Spatiotemporal Forecasting, NeurIPS, 2025.
>
> [2] Air-DualODE: Air Quality Prediction with Physics-Guided Dual Neural ODEs in Open Systems, ICLR, 2025.
>
> [3] TimeMixer: Decomposable Multiscale Mixing for Time Series Forecasting, ICLR 2024.
>
>
> ## [W1] Technical Novelty and Contribution
> We clarify that NeST's core contribution is a **novel macro-to-micro paradigm with incorporating future guidance**.
> Specifically, (1) rather than relying on predefined prior knowledge (e.g., static graphs), we leverage spectral clustering to group temporally correlated trajectories, enabling adaptive spatial abstraction; and (2) instead of a static hierarchy, our future guidance (via bi-directional attention) provides a consistent global signal for autoregressive modeling, effectively controlling the accumulative errors in long-term forecasting.
>
> ## [W2] Decoupled Offline Clustering
> We intentionally decoupled clustering from forecasting objective to prevent training instability and mode collapse (e.g., all nodes in one region) common in joint optimization.
> To construct the affinity matrix, we explicitly account for dynamical trends, defining distances based on temporal patch-wise differences, enabling the clustering to better capture underlying spatio-temporal patterns.
>
> ## [W3] Influence of Regional Dynamics
> In NeST, regional dynamics act as **soft conditions** via cross-scale attention. When a node aligns with its region, the model leverages this stable macro signal to reduce prediction drift.
> Conversely, during a local anomaly contradicting the macro trend, attention **dynamically downweights** macro guidance, relying more on local features. This allows the model to benefit from **regional stability** while adapting to **local variations**.
> We will include this analysis in the revised manuscript.

---

> > ### Author Rebuttal · Reviewer_dTqY · 2026-04-03
> >
> > Thanks for the authors' response.  I have read the rebuttal and slightly increased my scores.

---

> > > ### Author Response · Authors · 2026-04-03
> > >
> > > Dear Reviewer dTqY,
> > >
> > > Thank you very much for acknowledging our rebuttal. We are truly grateful for your confirmation and the positive support you have shown to us all the time! We noticed the score remains unchanged in the system. We wanted to gently mention this just in case it was a systematic issue. Regardless, we deeply appreciate your constructive suggestions and the time you have dedicated to our work. Thank you!
> > >
> > > Authors

---

### Decision · Program_Chairs · 2026-04-30

**Decision:**

Accept (regular)

**Comment:**

This paper proposed a nested forecasting framework which couples future macro-level regional trends with micro-level historical observations, enabling top-down guidance from abstract future representations for fine-grained forecasting. Reviewers praised its idea novelty, unification of existing methods, and solid experiments. During rebuttal, three out of four reviewers provided positive feedbacks on this paper. Therefore, I recommend acceptance of this paper.